# HOW TO ENABLE UNCERTAINTY ESTIMATION IN PROXIMAL POLICY OPTIMIZATION

## ABSTRACT

While deep reinforcement learning (RL) agents have showcased strong results across many domains, a major concern is their inherent opaqueness and the safety of such systems in real-world use cases. To resolve these issues, we need agents that can quantify their uncertainty and detect *out-of-distribution* (OOD) states. Existing uncertainty estimation techniques, like Monte-Carlo Dropout or Deep Ensembles, have not been extended to on-policy deep RL. We posit that this is due to two reasons: concepts like uncertainty and OOD states are not well defined compared to supervised learning, especially for on-policy RL methods. Secondly, available implementations and comparative studies for uncertainty estimation methods in RL have been limited. To fill the first gap, we propose definitions of uncertainty and OOD for Actor-Critic RL algorithms, namely, proximal policy optimization (PPO), and present potentially applicable measures. In particular, we discuss the concepts of value and policy uncertainty. The second problem is addressed by implementing different uncertainty estimation methods and comparing them across a number of environments. The OOD detection performance is evaluated via a custom evaluation benchmark of *in-distribution* (ID) and OOD states for various RL environments. We identify a trade-off between reward and OOD detection performance. To overcome this, we formulate a bi-objective meta optimization problem in which we simultaneously optimize hyperparameters for reward and OOD detection performance. Our experiments show that the recently proposed method of *Masksembles* (Durasov et al. (2021)) strikes a favourable balance among the methods, enabling high-quality uncertainty estimation and OOD detection that matches the performance of original RL agents.

## 1 INTRODUCTION

Agents trained via deep RL have achieved several high-profile successes over the last years, e.g., in domains like game playing (Badia et al. (2020)), robotics (Gu et al. (2017)), or navigation (Kahn et al. (2017)). However, generalization and robustness to different conditions and tasks remain a considerable challenge (Kirk et al. (2021); Lütjens et al. (2020)). Without specialized training procedures, e.g., large-scale multitask learning, RL agents only learn a very specific task in given environment conditions (OEL Team (2021). Moreover, even slight variations in the conditions compared to the training environment can lead to severe failure of the agent (Lütjens et al. (2019); Dulac-Arnold et al. (2019)). This is especially relevant for potential safety critical applications. Methods to uncover an agent's uncertainty can help to combat some of the most severe consequences of incorrect decision-making. Agents that could indicate high uncertainty and, therefore, e.g., query for human interventions or use a robust baseline policy (e.g., a hard-coded one) could vastly improve the reliability of systems deploying RL-trained decision-making.

In response, uncertainty estimation has been recognized as an important open issue in deep learning (Abdar et al. (2021); Malinin & Gales (2018)) and has been investigated in domains such as computer vision (Lakshminarayanan et al. (2017); Durasov et al. (2021)) and natural language processing (Xiao & Wang (2018); He et al. (2020)). The topic of uncertainty estimation for deep RL has also gained traction (Clements et al. (2019)). While some work has explored OOD detection for Q-learning-based algorithms (Chen et al. (2017; 2021); Lee et al. (2021)), there is almost a complete lack of studies on uncertainty estimation for actor-critic algorithms, despite the fact that actor-critic algorithms are widely used in practice, e.g., in continuous control applications. Additionally, work specifically

exploring OOD detection has been mostly focused on very simple environments, and neural network architectures (Mohammed & Valdenegro-Toro (2021); Sedlmeier et al. (2019)). This, therefore, limits the transferability of results to more commonly used research environments. In this paper, we bridge this research gap through a systematic study of *in-distribution (ID)* and *out-of-distribution (OOD)* detection for on-policy actor-critic RL across a set of more complex environments.

We implement and evaluate a set of uncertainty estimation methods for Proximal Policy Estimation (PPO) (Schulman et al. (2017)), a versatile on-policy actor-critic algorithm that is popular in research due to its simplicity and high performance. We compare a series of established uncertainty estimation methods in this setting. Specifically, we introduce the recently proposed method of *Masksembles* (Durasov et al. (2021)) to the domain of RL. For multi-sample uncertainty estimation methods like Ensembles or Dropout, we identify a trade-off in on-policy RL: During training, a "closeness" to the on-policy setting is required, i.e., models should train on their own recent trajectory data. However, for OOD detection, a sufficiently diverse set of sub-models is required, which leads to the training data becoming off-policy. Existing methods like Monte Carlo Dropout (MC Dropout), Monte Carlo Dropconnect (MC Dropconnect), or Ensembles, therefore, struggle either to achieve good reward or OOD detection performance. The recently proposed method of Masksembles has the ability to smoothly interpolate between different degrees of inter-model correlation via its hyperparameters. We propose a bi-objective meta optimization to find the ideal configuration. We show that Masksembles can produce competitive results for OOD detection while maintaining good performance.

The contributions of this paper are three-fold: (1) We examine the concept of uncertainty for on-policy actor-critic RL, discussing ID and OOD settings; to tackle this, we define value- and policy-uncertainty, as well as a multiplicative measure (Sec. 3). (2) We show different methods to enable uncertainty estimation in PPO; as part of it, we establish Masksembles as a robust method in the domain of RL, particularly for on-policy actor-critic RL (Sec. 4). (3) We present an OOD detection benchmark to evaluate the quality of uncertainty estimation for a series of MuJoCo and Atari environments (Sec. 5), we evaluate the presented methods and highlight key results of the comparison (Sec. 6).
The full source-code of the implementations, fully compatible with *StableBaselines3* (Raffin et al. (2021)), will be released as open-source.

## 2 RELATED WORK

**Defining uncertainty in RL** Defining uncertainty for RL is less well studied compared to the supervised setting (Abdar et al. (2021)). Existing work studied aleatoric and epistemic uncertainty in reinforcement learning (Clements et al. (2019); Charpentier et al. (2022)), e.g., disentangling the two types of uncertainty. Here, we are interested in OOD detection during inference which is achieved by capturing epistemic uncertainty. However, as we do not directly attempt to disentangle both sources, our proposed uncertainty measures can also be interpreted as estimating total uncertainty. **Methods for uncertainty estimation in RL** Different methods have been proposed in supervised learning settings to tackle uncertainty estimation and OOD detection. Bayesian deep learning (Kendall & Gal (2017); Wang & Yeung (2020)) methods provide robust uncertainty estimation rooted in deep theoretical background. However, Bayesian deep learning methods often come with a considerable overhead in complexity, computation, and sample inefficiency compared to non-Bayesian methods. This has slowed their adoption in deep learning in general and in RL in particular. Deep ensemble models have proven a reliable and scalable method for uncertainty and OOD detection in supervised deep learning (Lakshminarayanan et al. (2017); Fort et al. (2019)). An obvious drawback of ensemble methods is the need for a number of independently trained models, which drives up computational cost and inference time. MC Dropout has been proposed as a more resource-effective method for single-model uncertainty estimation (Gal & Ghahramani (2016)). As a disadvantage, enabling drop-out during inference can lead to a lower model performance overall (Durasov et al. (2021)). Although there has been work in the space of using uncertainty estimation in RL, it has mostly been limited to either simple bandit settings (Auer (2003); Lu & Van Roy (2017)), model-based RL (Kahn et al. (2017)), or applied in training routines to improve the agent performance, e.g. by boosting exploration in uncertain states (Wu et al. (2021); Chen et al. (2017); Osband et al. (2016; 2018)). Work explicitly investigating OOD detection for RL has been limited (Mohammed & Valdenegro-Toro (2021); Sedlmeier et al. (2019)). Out of the existing work, most methods have been almost exclusively applied to Q-learning (Mohammed & Valdenegro-Toro (2021); Sedlmeier et al. (2019);

Chen et al. (2017; 2021); Lee et al. (2021)). In contrast, our work specifically targets for on-policy and actor-critic methods, including for environments with continuous action spaces.

**Evaluating uncertainty estimation in RL** Work on the systematic evaluation and benchmarking of uncertainty estimation or OOD detection methods in RL has been limited. (Sedlmeier et al. (2019)) propose a very simple environment based on a gridworld setting and flipping on the goal state to benchmark a set of uncertainty estimation methods. (Mohammed & Valdenegro-Toro (2021)) introduce physics-based interventions for simple control tasks like Cartpole and compare some methods. We extend these interventions to more complex simulation-based environments (Todorov et al. (2012)). Additionally, the existing work has a very limited discussion on reward performance, training details, or used algorithms. In this work, we keep a stronger focus on implementation, the relationship between reward and OOD detection performance, as well as effect of hyperparameters.

## 3    UNCERTAINTY FOR ON-POLICY ACTOR-CRITIC REINFORCEMENT LEARNING

In this section, we provide the background on actor-critic RL. We first discuss the concept of uncertainty as well as ID and OOD data for on-policy algorithms. We then introduce value uncertainty and policy uncertainty to address the architecture of actor-critic algorithms.

**Actor-Critic Reinforcement Learning** *Markov decision process* is defined as $\mathcal{M} = (\mathcal{S}, \mathcal{A}, \mathcal{P}, R, P_0, \gamma)$, where $\mathcal{S}$ is the state space, $\mathcal{A}$ is the action space, $\mathcal{P}$ is the family of transition distributions on $\mathcal{S}$ indexed by $\mathcal{S} \times \mathcal{A}$ with $p(s'|s, a)$ describing the probability of transitioning to state $s'$ when taking action $a$ in state $s$, $R : \mathcal{S} \times \mathcal{A} \to \mathbb{R}$ is the reward function, $P_0$ is the initial state distribution, and $\gamma$ is the discount factor.

A *policy* $\pi_\theta$ is a conditional probability distribution of actions $a \in \mathcal{A}$ given states $s \in \mathcal{S}$ parameterized by $\theta$. RL is the problem of finding the optimal policy of the agent with respect to the specified reward function. In this work, we investigate uncertainty estimation for an actor-critic algorithm. Formally, the actor/policy $\pi_\theta(a|s_t)$ outputs a probability to take an action $a \in \mathcal{A}$ in a state $s$. In actor-critic algorithms, we can define the objective of the policy and the definition of the critic/value function:

$$L_{\text{act.}}(\theta) = \hat{\mathbb{E}}[\log \pi_\theta(a_t|s_t)\hat{A}_t] \quad \hat{V}_\psi^\pi(s_t) = \hat{\mathbb{E}}[\sum_{l=0}^{\infty} \gamma^l R(s_{t+l})]. \tag{1}$$

In line with common PPO implementations, throughout the paper, we assume the advantage function $\hat{A}_t$ to be computed via generalized advantage estimation (GAE) (Schulman et al. (2016)), which internally uses the value function estimator $\hat{V}_\psi^\pi(s)$ associated with policy $\pi$ and parameterized by $\psi$. Further in the text, we will use the notation $\hat{V}$ for simplicity. Both the policy $\pi$ and value estimator $\hat{V}$ are implemented as a neural network. In the on-policy case, the trajectories to update both the policy and value function are rollouts from the most recent policy $\pi_\theta$. For off-policy algorithms, accumulated data from current and past policy rollouts are also utilized for optimization. While this re-sampling increases sample efficiency, it is traded with training instability. In on-policy RL, the target policy is directly optimized to maximize cumulative reward.

### 3.1    DEFINING IN-DISTRIBUTION AND OUT-OF-DISTRIBUTION FOR ON-POLICY RL

Common implementations of actor-critic variants lack a way to output uncertainty with respect to states. We are interested in quantifying uncertainty to find OOD states for an RL agent. In supervised learning, there is a comparatively clear distinction between ID and OOD cases: data that is similar to the training distribution can be generally considered as in-distribution. Importantly, the training distribution is known and fixed at training time. OOD data generally describes data that is not part of the learned manifold describing the training distribution, which means, e.g., a classifier is unable to interpolate to an OOD sample.

The definition of OOD for Markovian data is more complicated than the ones for the independent identically distributed (i.i.d.) case because the training data is constantly changing over the training process. Fig. 1 shows a schematic explanation of ID for different types of RL algorithms. Depending on the type of RL, offline-, off-policy, or on-policy setting, we assume a different set of states to be *in-distribution* or *out-of-distribution*.

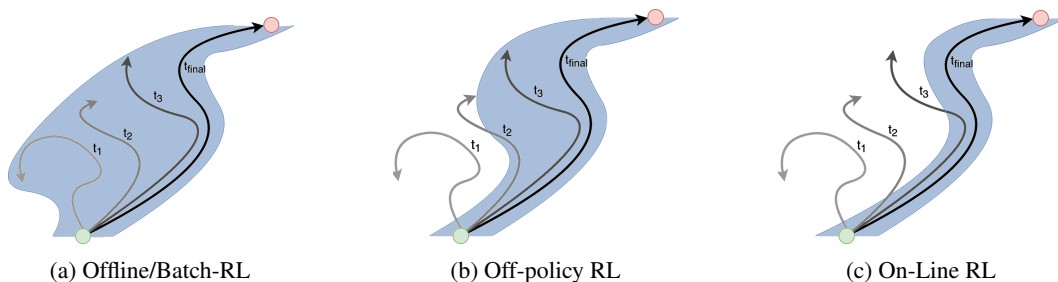

| (a) Offline/Batch-RL | (b) Off-policy RL | (c) On-Line RL |

Figure 1: The RL setting influences what can be considered ID and OOD. Depending on the setting, a different area of the state space can be considered ID for value estimation. We use the background cones to depict the part of state space that is considered as ID; $t_1, t_2, t_3, t_{final}$ to notate schematic trajectories which have evolved with training time. The green dot shows the initial state of the agent, and the red one is the terminal state.

Because an offline-RL agent is repeatedly trained on a full batch of optimal and suboptimal trajectories, it can produce consistent estimates even for these suboptimal states. In off-policy settings, where a large replay buffer is used, compare, e.g., to DQN (Mnih et al. (2015)) or SAC (Haarnoja et al. (2018)), we have a combination of current and outdated states, i.e., the agent is still trained to predict values for states which are not part of the trajectories produced by the most recent policies. In the on-policy case, we assume that only a narrow set of states along the trajectories produced by active policy can be considered as ID. This is because both the value estimator and the policy are only updated on data from the active policy. As part of the policy optimization procedure, due to *catastrophic forgetting* (French (1999)), states encountered early during training but not visited frequently due to the policy shifting to other trajectories cannot be considered ID anymore. Importantly, this also means that the set of ID states is specific to the trained policy. Our goal is to enable the trained policy to detect OOD states at inference time. Because of the aforementioned "closeness" of both policy and value function, we will investigate the suitability of both, the uncertainty of the value function, and the uncertainty of the policy function to quantify the agent's uncertainty.

### 3.2 QUANTIFYING UNCERTAINTY: POLICY UNCERTAINTY AND VALUE UNCERTAINTY

We now discuss possible uncertainty measures for actor-critic RL. As mentioned, our goal is to quantify the agent's uncertainty, i.e., the uncertainty of decision-making. A direct application of this is the detection of OOD states in which the agent is not able to predict an action to maximize return with high confidence. In actor-critic RL, we can apply sample-based uncertainty estimation methods to both the policy and value network, which means we both get an output distribution for the action and the value estimate. This enables us to investigate both policy and value uncertainty independently to probe their adequateness as uncertainty measures.

During the remainder of the paper, we use the term **policy uncertainty** to refer to any uncertainty measure for the output distribution of the policy. In the following evaluation, we encounter both categorical and continuous output distributions. Formally they are defined as:

$$\pi_{\text{cat}}(a_i|s;\theta) = \frac{\exp(\tilde{\pi}(a_i|s;\theta))}{\sum_{j=0}^{N}\exp(\tilde{\pi}(a_j|s;\theta))} \qquad \pi_{\text{con}}(a|s;\theta) \sim \mathcal{N}_{j=0..N}(\tilde{\pi}(a^j|s), \sigma_{fix}) \qquad (2)$$

where $a_i$ is the $i$-th action of a discrete set of actions, and $a^j$ is the $j$-th component of a multi-variate continuous action. In the case of discrete actions, $N$, therefore, refers to the number of actions, and in the case of continuous, $N$ refers to the dimension of the multi-variate action. For discrete actions, a categorical distribution is defined by a softmax of the logits $\tilde{\pi}$, i.e., the output of the policy network. For continuous actions, a multi-variate diagonal Gaussian distribution is applied, with the mean of each dimension parameterized by the logits of the policy network and a state-independent $\sigma$ (reported in Appendix A).

In the following, we use multiple predictions of different sub-models during inference, we write $\Pi = \{\pi^1, \pi^2, ..., \pi^k\}$ for $k$ different sub-models/samples. From the perspective of supervised classification, a possible uncertainty measure is $\mathcal{U}_{cat,max.prob.}^{\Pi}(s) = 1 - p_{max}(s) = 1 - \max_{j\in 1,..,N} \pi_\mu(a^j|s)$

of the averaged distribution $\pi_\mu(a|s) = \frac{1}{k}\sum_{h=0}^{k}\pi^h(a|s)$. This corresponds to the confidence with which the classifier assigns the top label. However, in RL, there is no clear singular correct action, and thus a low probability for a certain action does not necessarily correspond to high uncertainty. It could also mean that two actions are equivalent with respect to the optimal policy. Thus, instead of capturing epistemic uncertainty, which is required for OOD detection, the measure instead captures the irreducible uncertainty of the environment dynamics, i.e. aleatoric uncertainty. For the same reason, the entropy of the averaged distribution $\mathcal{U}_{cat,entropy}^{\Pi}(s) = H(\pi_\mu(a|s))$ also does not serve as a reliable uncertainty measure. Lastly, both of these measures are influenced by the magnitude of an entropy bonus to the policy optimization objective (see (Schulman et al. (2017)) for details), which further limits general applicability. In Sec. 6, we report the performance of both uncertainty measures but found them to be unreliable for quantifying uncertainty w.r.t. ID or OOD states.

Instead, we focused on uncertainty measures that are based purely on the disagreement between the respective sub-models. We use the following formulations for policy uncertainty for continuous and categorical actions, respectively:

$$\mathcal{U}_{con,std}^{\Pi}(s) = \frac{1}{N}\sum_{j=0}^{N} std(\{\pi^1(a^j|s), \pi^2(a^j|s), ..., \pi^k(a^j|s)\}) \qquad (3)$$

and

$$\mathcal{U}_{cat,std}^{\Pi}(s) = \frac{1}{N}\sum_{i=0}^{N} std(\{\pi^1(a_i|s), \pi^2(a_i|s), ..., \pi^k(a_i|s)\}). \qquad (4)$$

We chose this formulation, because both $\pi(a_i|s)$ and $\pi(a^j|s)$ are directly parameterized by the logits of the respective policy networks. We, therefore, hypothesize that we can interpret these average standard deviations as a measure of the agreement/disagreement between the sub-policies $\pi^1, .., \pi^k$. An alternative approaches to formulate uncertainty can be found in Appendix E.

Equivalently, we refer to uncertainty measures for the value function as **value uncertainty** and use the following notation $\mathcal{U}^{\mathcal{V}}$ with $\mathcal{V} = \{\hat{V}^1, \hat{V}^2, .., \hat{V}^k\}$. We investigate the standard deviation of the value function as a potential proxy for state-visitation frequency:

$$\mathcal{U}_{std}^{\mathcal{V}}(s) = std(\{\hat{V}^1(s), \hat{V}^2(s), ..., \hat{V}^k(s)\}). \qquad (5)$$

Based on these measures, we define **multiplicative uncertainty**, which we define as the product between policy and value uncertainty for each state. We may interpret this measure as a state-action uncertainty measure:

$$\mathcal{U}_{std}^{Mult}(s) = \mathcal{U}_{std}^{\Pi}(s) * \mathcal{U}_{std}^{\mathcal{V}}(s). \qquad (6)$$

## 4 EQUIPPING PPO WITH UNCERTAINTY ESTIMATION

Based on the defined uncertainty metrics, we now investigate the integration of uncertainty estimation methods into RL actor-critic architectures. We discuss how to implement uncertainty estimation for an actor-critic algorithm, specifically PPO (Schulman et al. (2017)).

### 4.1 INTEGRATING UNCERTAINTY ESTIMATION

As a pre-condition, we want the uncertainty methods to be easily integrated into existing RL setups as drop-in replacements for existing network architectures. While this limits the set of methods, it ensures high practical applicability of the presented results.

Recently, Masksembles (Durasov et al. (2021)) were proposed as an attempt to combine the advantages of Ensembles and MC Dropout: MC Dropout randomly discards some parts of the network by applying a random binary mask at every iteration, i.e., using an infinite number of overlapping models. Ensembles use a set of fixed and separate sub-models. Masksembles create only a limited and fixed set of masks but also share part of the network. Importantly, Masksembles provides a parameter to control the overlap between masks, which in turn controls the correlation between the submodels. This allows us to flexibly determine the best configuration based on the problem. In terms of training speed, Masksembles performs comparably to MC Dropout and outperforms the latter in terms of

inference speed because Masksembles can generate output distributions in a single forward-pass. While the technique was proposed in the domain of image classification, to our knowledge, this is the first use for RL.

We compare our Masksembles-based model (Fig. 2e) against three architectures enabling uncertainty estimation and baseline. Both MC Dropout and Ensembles are established uncertainty estimation methods. Additionally, we compare against the more recent MC Dropconnect. Fig. 2 is a schematic illustration of the integration of different methods in the baseline MLP architecture used for continuous control tasks. Architecture details are discussed in Appendix A.

**Baseline:** The default fully-connected networks and CNNs serve as a comparison for reward performance. It does not provide uncertainty estimation functionality (Fig. 2a).

**Masksembles (Durasov et al. (2021)):** During training and inference, masks are applied across the input batch during the forward pass, which leads to $k$ different predictions used for uncertainty estimation. We base our design choices on the respective paper (Fig. 2e).

**Monte-Carlo Dropout (Gal & Ghahramani (2016)):** Here, we train the networks with Dropout enabled. We apply Dropout to the same two layers as Masksembles to enable direct comparability. For MC Dropout, we keep Dropout enabled during inference and sample multiple times with individual Dropout masks to get a distribution of predictions. Compared to Masksembles, Dropout masks are unique for each prediction. To match the number of masks in Masksembles, we base the uncertainty estimation on $k = 4$ independent samples (Fig. 2b).

**Monte-Carlo Dropconnect (Mobiny et al. (2021)):** Dropconnect is closely related to Dropout. Instead of disabling neurons, i.e., the output of a layer, random weights are assigned the value 0, which achieves a similar effect as Dropout (Fig. 2c).

**Ensembles:** Lastly, as a comparative baseline, we employ an ensemble of 4 networks, in the case of MuJoCo 4 policy and critic networks each, in the case of Atari 4 independent CNNs with individual policy and value heads each. As mentioned, to ensure basic synchronization between the networks, all networks are trained based on the loss function computed from a shared replay buffer (Fig. 2d).

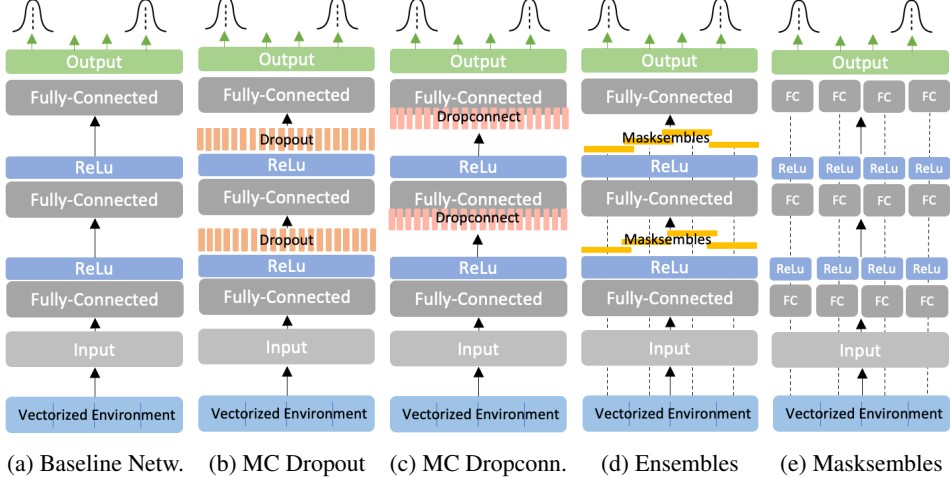

(a) Baseline Netw.  (b) MC Dropout  (c) MC Dropconn.  (d) Ensembles  (e) Masksembles

Figure 2: Different MLP architectures: The baseline network (a) is a three-layer MLP. For MC Dropout (b), a random mask is applied to each prediction, which can be interpreted as an infinite number of sub-models. (c) Dropconnect is similar to Dropout. For Ensembles (d) we assume a small number of fully independent networks. Masksembles (e) can be interpreted as an interpolation that uses a fixed set of masks to create partially overlapping sub-models. For Masksembles and Ensembles, each sub-model/mask is assigned to a fixed subset of environments (dashed lines).

## 4.2 INFERENCE WITH MULTI-SAMPLE METHODS

We apply multiple inference schemes to translate the distribution of policy sub-models into a final action prediction. For categorical actions, in the games of *SpaceInvaders* and *MSsPacman*, we use a voting scheme, which is commonly applied in classification: Each of the sub-model assigns probabilities to the available actions. We then choose the action with the highest probability from each submodel and then sample the mode of these $k$ predicted actions. In case two or more actions

are chosen in an equal amount, we randomly decide on one of the actions. In preliminary experiments, we tried averaging the $k$ categorical action distributions across actions and then sampling from the averaged distribution, and found this to have slightly better performance in *Berzerk* and *SpaceInvaders*. For each of the games, we report the best performing configuration. For continuous actions, we use the submodel predictions to parameterize a diagonal Gaussian distribution. For deterministic action selection, we choose the mean predictions; for stochastic selection, we sample from the distributions.

## 5 BENCHMARKING OUT-OF-DISTRIBUTION STATE DETECTION

**Environments** We evaluate the presented methods on four MuJoCo (Todorov et al. (2012)) control environments, namely, Ant-v3, HalfCheetah-v3, Swimmer-v3, and Walker2d-v3. All of these environments feature vector-valued continuous actions. Furthermore, we apply the surveyed methods to four Atari environments (Bellemare et al. (2013)): Seaquest-v4, MsPacman-v4, Berzerk-v4 and SpaceInvaders-v4. We chose these four games to represent different game-play mechanics and because recorded game-play is available as part of the Atari-Head dataset ((Zhang et al. (2019)), which we use for OOD state generation as described below.

**Creating the Benchmark for Out-of-distribution Data** For MuJoCo-based environments, we apply a number of random perturbations to the parameters of the physics simulation, e.g., increasing/decreasing gravity and/or friction or varying the dimensions of the body elements. These modifications create OOD states when the agent acts in the environment. In Appendix H, we give a detailed description of the applied interventions. To create OOD states, we randomly sample from the set of available parameters and let the respective agent act for 100 steps in the perturbed environment; 50 configurations are sampled to create a dataset of possible OOD states.
For Atari games, previous work has proposed image-based perturbations (like Gaussian noise, blur, etc.) to create OOD states (Mohammed & Valdenegro-Toro (2021)). While this is a simple and generalizable method to create these states, we feel that this approach does not necessarily capture the nature of OOD states in Markovian settings. Therefore, we focus on valid game states that the agent has not yet encountered. Specifically, we use sampled trajectories of advanced levels from the Atari-HEAD dataset (Zhang et al. (2019)). We ensure that the sampled states are from states of the games that the trained agents have not yet reached due to insufficient performance (e.g., with a changed level layout). We give examples of OOD states in the Atari setting in Appendix H. We report some additional results for additional image-based perturbation techniques, like adding noise or masking parts of the input image, in Appendix I.

**Methodology for Evaluation** We sample ID data directly by executing the trained policies in unperturbed environments. This is connected to the discussion of Sec. 3, that ID states are densely distributed around the final model trajectories. OOD states are created as described above. An effective uncertainty measure should be substantially higher for OOD states compared to ID states. We frame OOD detection as a binary classification problem: We assign ID and OOD respective labels. Then, the computed uncertainty scores, as described in Sec. 3, are used as a threshold to classify ID and OOD states. We then compute ROC-AUC to evaluate the OOD detection quality.

**Hyperparameter Optimization** Most of the results that will be reported in Sec. 6 are achieved with hyper-parameters that were optimized for the baseline algorithm, i.e., without specifically fine-tuning the specific methods. This means that, in general, the reported results are representative of using the presented methods as full drop-in replacements without any modifications to the training process. However, we are interested in uncertainty estimation methods that are robust to the choice of hyper-parameters (like a number of models, learning rate, batch size, etc.). To enable a systematic investigation of the sensitivity to hyper-parameter choices, we run a bi-objective meta optimization procedure: We are interested in both achieving a high cumulative episode reward and a strong OOD detection performance. We sample 190 configurations per method and environment using *Optuna* (Akiba et al. (2019)). During the optimization, we choose a simplified OOD state generation scheme: We add uniform noise with a magnitude of $2\times$ the element-wise standard deviation to the input observations in order to create OOD states. We chose this scheme due to its efficiency in large-scale meta optimization. We see this generation schema as supplemental to the OOD generation method described above. We showcase the results of this procedure in Fig. 4. As we have mentioned previously, this generation schema is not suitable to reliably evaluate epistemic uncertainty, it gives an approximimation of the methods capabilities.

| Method | HalfCheetah-v3 | Ant-v3 | Walker2d-v3 | Swimmer-v3 | Seaquest-v4 | MsPacman-v4 | Berzerk-v4 | SpaceInvaders-v4 |
|---|---|---|---|---|---|---|---|---|
| Baseline | $3056.5_{\pm695.9}$ | $5189.7_{\pm661.7}$ | $2958.6_{\pm1529.3}$ | $178.8_{\pm123.7}$ | $3898.6_{\pm404.5}$ | $2405.0_{\pm690.2}$ | $605.3_{\pm382.8}$ | $865.6_{\pm167.4}$ |
| MS | $\mathbf{2673.6}_{\pm556.6}$ | $\mathbf{4884.8}_{\pm926.9}$ | $\mathbf{2328.2}_{\pm480.5}$ | $\mathbf{151.4}_{\pm63.4}$ | $3524.0_{\pm1230.3}$ | $1855.6_{\pm876.8}$ | $\mathbf{761.3}_{\pm233.1}$ | $587.3_{\pm33.4}$ |
| MCDO | $2014.6_{\pm529.6}$ | $2471.3_{\pm512.2}$ | $656.7_{\pm121.7}$ | $78.2_{\pm7.8}$ | $\mathbf{3445.3}_{\pm1154.7}$ | $\mathbf{2169.3}_{\pm27.3}$ | $562.3_{\pm154.1}$ | $706.6_{\pm154.5}$ |
| MCDC | $651.8_{\pm560.0}$ | $1610.6_{\pm542.7}$ | $859.4_{\pm68.6}$ | $49.3_{\pm13.1}$ | $3110.3_{\pm972.1}$ | $1696.6_{\pm703.9}$ | $637.6_{\pm160.1}$ | $\mathbf{759.5}_{\pm60.4}$ |
| Ensembles | $713.4_{\pm283.6}$ | $989.2_{\pm37.4}$ | $128.9_{\pm42.3}$ | $26.4_{\pm6.3}$ | $122.6_{\pm42.1}$ | $210.0_{\pm14.0}$ | $460.2_{\pm277.6}$ | $270.0_{\pm0.0}$ |
| MS Single | $\mathbf{2654.9}_{\pm490.7}$ | $5746.3_{\pm830.4}$ | $\mathbf{2007.1}_{\pm760.3}$ | $\mathbf{172.6}_{\pm52.4}$ | $3242.6_{\pm1107.3}$ | $1813.6_{\pm851.7}$ | $\mathbf{708.0}_{\pm133.1}$ | $\mathbf{774.2}_{\pm71.0}$ |
| MCDO Single | $2008.8_{\pm541.5}$ | $2272.4_{\pm958.3}$ | $669.9_{\pm192.1}$ | $111.5_{\pm21.0}$ | $\mathbf{3540.6}_{\pm1229.7}$ | $\mathbf{2204.3}_{\pm38.5}$ | $632.6_{\pm108.7}$ | $583.3_{\pm108.7}$ |
| MCDC Single | $1006.6_{\pm689.3}$ | $2384.5_{\pm906.3}$ | $671.0_{\pm275.2}$ | $58.4_{\pm17.0}$ | $3186.0_{\pm1034.4}$ | $1681.6_{\pm713.7}$ | $675.3_{\pm95.5}$ | $780.3_{\pm60.4}$ |
| Ens. Single | $464.0_{\pm355.7}$ | $989.3_{\pm4.4}$ | $128.5_{\pm10.3}$ | $14.6_{\pm3.3}$ | $183.3_{\pm127.9}$ | $247.6_{\pm53.2}$ | $283.6_{\pm60.1}$ | $268.3_{\pm50.9}$ |

Table 1: Reward summary across environments and methods Masksembles (MS), Monte Carlo Dropout (MCDO), Monte Carlo Dropconnect (MCDC) and Ensembles. Rewards are measured over 10 episodes and averaged over 3 seeds. The upper results are for action sampling based on model averaging/voting. The *Single* scores are for action prediction with a single random sub-model. For *Berzerk-v4*, we truncated episodes after 10.000 steps if case an agent gets stuck. We mark the best results for every environment in **bold**.

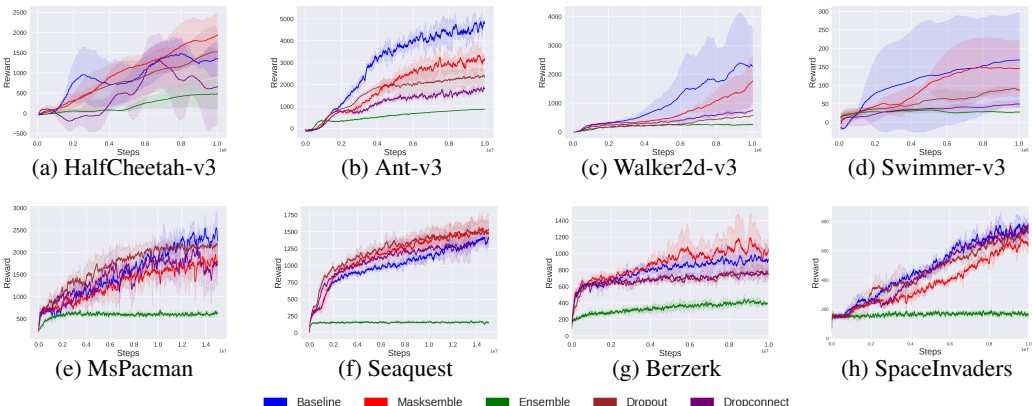

|  |  |  |  |
|---|---|---|---|
| (a) HalfCheetah-v3 | (b) Ant-v3 | (c) Walker2d-v3 | (d) Swimmer-v3 |
| (e) MsPacman | (f) Seaquest | (g) Berzerk | (h) SpaceInvaders |

Baseline — Masksemble — Ensemble — Dropout — Dropconnect

Figure 3: First Row: Training Curves for *MuJoCo*; Second Row: Training Curves for *Atari*.

# 6 EVALUATION RESULTS

In this section, we focus on comparing Masksembles approach following the protocol described in Sec. 5, namely, we measure training results, the performance of the agents based on the ground-truth reward by rolling out trained policies, and the ability to detect OOD states.

**Reward** Following the procedure defined in Sec. A.1 we observe the training dynamic for control-problems depicted in Fig. 3b, 3a, 3c, 3d and games depicted in Fig. 3e,3f,3c, 3h. During training, actions are only selected via single samples, i.e., one prediction per environment. During inference, multiple samples can be combined into a single prediction as described in Sec. 4.2. The results are presented in Tab. 1. We also report the rewards achieved via inference with a single model. From the results, we cannot conclude that combining multiple predictions via voting or averaging leads to a higher reward compared to a single model.

**Out-of-distribution Detection** The results for the previously described OOD detection benchmark (see Sec. 5) are summarized in Tab. 2. We found that in most cases, the Masksembles approach outperformed MC Dropout, MC Dropconnect, and Ensembles in terms of OOD detection. Interestingly, in the control environment, e.g., Ant-v3, value uncertainty also acts as a measure of physical instability, particularly detachment from the floor, e.g., it explains substantial uncertainty at the beginning of an episode when Ant is spawned. Such "levitation" states mean that any action can be equivalently reasonable or impaired concerning expected discounted reward.

**Bi-Objective Meta Optimization** The results of the hyperparameter optimization are visualized in Appendix B. The results show that Masksembles has a favorable distribution of results compared to the other methods. We find that both in terms of reward and specifically in terms of ROC-AUC score for OOD detection, Masksembles has a larger set of well performing configurations. We also find that the best results for Masksembles are often close to the baseline. Some agents trained with MC Dropout and MC Dropconnect also show a good performance, but the number of good configurations

| Method | H.Cheetah-v3 | Ant-v3 | Walker2d-v3 | Swimmer-v3 | MsPacman-v4 | Seaquest-v4 | Berzerk-v4 | SpaceInv.-v4 |
|---|---|---|---|---|---|---|---|---|
| MS V. U. (ours) | $\mathbf{0.85}_{\pm 0.067}$ | $\mathbf{0.73}_{\pm 0.139}$ | $\mathbf{0.73}_{\pm 00.067}$ | $0.88_{\pm 0.075}$ | $0.89_{\pm 0.062}$ | $\mathbf{0.82}_{\pm 0.067}$ | $\mathbf{0.73}_{\pm 0.176}$ | $0.57_{\pm 0.011}$ |
| MCDO V.U. | $0.47_{\pm 0.143}$ | $0.34_{\pm 0.088}$ | $0.48_{\pm 0.043}$ | $0.87_{\pm 0.043}$ | $\mathbf{0.95}_{\pm 0.012}$ | $0.75_{\pm 0.022}$ | $0.6_{\pm 0.016}$ | $0.58_{\pm 0.009}$ |
| MCDC V.U. | $0.32_{\pm 0.07}$ | $0.65_{\pm 0.024}$ | $0.52_{\pm 0.034}$ | $0.44_{\pm 0.127}$ | $0.74_{\pm 0.097}$ | $0.69_{\pm 0.027}$ | $0.59_{\pm 0.097}$ | $\mathbf{0.61}_{\pm 0.068}$ |
| Ensemble V.U. | $0.74_{\pm 0.104}$ | $0.56_{\pm 0.012}$ | $0.63_{\pm 0.037}$ | $0.56_{\pm 0.037}$ | $\mathbf{0.94}_{\pm 0.039}$ | $0.52_{\pm 0.004}$ | $0.63_{\pm 0.025}$ | $0.5_{\pm 0.013}$ |
| MS P. U. (ours) | $0.85_{\pm 0.072}$ | $\mathbf{0.78}_{\pm 0.155}$ | $0.6_{\pm 0.038}$ | $\mathbf{0.85}_{\pm 0.093}$ | $0.90_{\pm 0.013}$ | $\mathbf{0.78}_{\pm 0.027}$ | $0.67_{\pm 0.021}$ | $\mathbf{0.61}_{\pm 0.032}$ |
| MCDO P.U. | $0.32_{\pm 0.049}$ | $0.43_{\pm 0.071}$ | $0.49_{\pm 0.042}$ | $0.61_{\pm 0.037}$ | $\mathbf{0.94}_{\pm 0.013}$ | $0.72_{\pm 0.023}$ | $0.68_{\pm 0.083}$ | $0.59_{\pm 0.035}$ |
| MCDC P.U. | $0.54_{\pm 0.01}$ | $0.52_{\pm 0.01}$ | $0.53_{\pm 0.034}$ | $0.54_{\pm 0.025}$ | $0.73_{\pm 0.066}$ | $0.68_{\pm 0.012}$ | $\mathbf{0.69}_{\pm 0.183}$ | $0.52_{\pm 0.028}$ |
| Ensemble P.U. | $\mathbf{0.86}_{\pm 0.069}$ | $0.64_{\pm 0.047}$ | $\mathbf{0.72}_{\pm 0.023}$ | $0.85_{\pm 0.117}$ | $0.34_{\pm 0.016}$ | $0.51_{\pm 0.012}$ | $0.46_{\pm 0.041}$ | $0.47_{\pm 0.012}$ |
| MS Mult.U. | $\mathbf{0.87}_{\pm 0.032}$ | $\mathbf{0.77}_{\pm 0.13}$ | $\mathbf{0.72}_{\pm 0.019}$ | $0.9_{\pm 0.027}$ | $0.93_{\pm 0.018}$ | $\mathbf{0.85}_{\pm 0.033}$ | $\mathbf{0.78}_{\pm 0.122}$ | $\mathbf{0.62}_{\pm 0.028}$ |
| MCDO Mult.U. | $0.37_{\pm 0.066}$ | $0.37_{\pm 0.046}$ | $0.48_{\pm 0.033}$ | $0.85_{\pm 0.025}$ | $\mathbf{0.97}_{\pm 0.005}$ | $0.78_{\pm 0.01}$ | $0.67_{\pm 0.054}$ | $0.61_{\pm 0.02}$ |
| MCDC Mult.U. | $0.35_{\pm 0.025}$ | $0.64_{\pm 0.017}$ | $0.53_{\pm 0.034}$ | $0.47_{\pm 0.114}$ | $0.77_{\pm 0.088}$ | $0.73_{\pm 0.009}$ | $0.69_{\pm 0.072}$ | $0.59_{\pm 0.054}$ |
| Ensemble Mult.U. | $0.83_{\pm 0.049}$ | $0.61_{\pm 0.01}$ | $0.69_{\pm 0.01}$ | $\mathbf{0.94}_{\pm 0.013}$ | $0.51_{\pm 0.033}$ | $0.52_{\pm 0.004}$ | $0.52_{\pm 0.012}$ | $0.49_{\pm 0.006}$ |
| Max. Prob. P.U. (best) | N/A | N/A | N/A | N/A | 0.53 | 0.49 | 0.51 | 0.52 |
| Entropy P.U. (best) | N/A | N/A | N/A | N/A | 0.51 | 0.53 | 0.48 | 0.49 |

Table 2: ROC-AUC scores for OOD detection across environments and methods: MS, MC DO, MC DC and Ensembles. We mark the best methods both for value uncertainty (V.U.) and policy uncertainty (P.U.) in **bold**. For Max. Prob. and Entropy P.U. we present the single best results of any method. *For continuous actions, the uncertainty scores $\mathcal{U}_{max.prob.}^{\Pi}$ and $\mathcal{U}_{entropy}^{\Pi}$ is not applicable.*

is smaller, and there is a noticeable gap in OOD detection performance. Ensemble-based models show a very weak ability to learn good policies but can reach good OOD detection performance.

## 7 DISCUSSION AND CONCLUSION

In this paper, we investigated uncertainty estimation for RL. We first discussed the background of OOD detection and uncertainty for on-policy RL. We then introduced the concepts of value and policy uncertainty for actor-critic algorithms, leading to a new multiplicative measure. Masksembles was presented as a simple, reliable uncertainty estimation method for RL. We provided implementation details and described training procedure and inference with estimated uncertainty. Finally, we proposed a uncertainty evaluation benchmark and compared multiple methods.

In this work, we have focused on PPO as the algorithm of choice. We found that the value uncertainty measure can produces reliable estimates for OOD detection. While policy uncertainty does not produce notable performance by itself, the multiplicative measure outperforms value uncertainty in most cases. Value uncertainty captures the state uncertainty, while policy uncertainty could also be seen as a representation of action-uncertainty. Therefore, we might interpret the multiplicative measure as capturing state-action-uncertainty, similar to using Q-value uncertainty in previous approaches.

A second noticeable result of our evaluation is the weakness of ensemble-based approaches. We want to note that this is currently specific to PPO, as ensemble-based approaches have shown greater success in off-policy and DQN-based methods (Lee et al. (2021); Hiraoka et al. (2021)). In Appendix C.1 we provide an interpretation of these results. We hypothesize that samples from different models have a negative impact in on-policy RL, because they potentially introduce off-policy states during training. This is also the case for Dropout and Dropconnect, which introduce noise during training. In Appendix C.2 we present results for fully separate Ensembles. Because the data distribution can change significantly during training, an ensemble of independent, non-synchronized policies may diverge. This makes averaging of predictions challenging, and because the sub-models do not share a common training distribution, the definition of epistemic uncertainty becomes very challenging.

There are still limitations with the current work: We observe a certain drop-off in performance in some environments. The meta-optimization shows that particular runs can already achieve very competetive results. Specifically, we find specific benefits for Masksembles in increasing the MLP layer sizes; see Appendix F. This can offset the smaller layer size introduced by the fixed masks. However, we see the nessecity to further close the performance gap compared to the baseline architectures.

Future work will include the following conceptual avenues: (1) utilize uncertainty measures to improve the performance and exploration abilities of current on-policy RL algorithms, (2) use OOD detection to query for human input in high-uncertainty situations (3) discover a more elaborate way to define uncertainty and disentangle policy and value uncertainty. (4) compare our findings with results from additional algorithms and environments, for example off-policy-algorithms and additional control tasks. We report some initial results for other algorithms in Appendix D.

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

# A  TRAINING DETAILS

## A.1  ARCHITECTURE DETAILS & TRAINING

**Architecture**  We choose three-layer MLPs as a baseline architecture for continuous control, with fully independent networks for the policy and value function. We include *Masksembles*-layers as a drop-in after the first and second hidden layer of the MLP (see Fig 2 for an overview). For Atari, we apply common pre-processing techniques such as no-op actions at the beginning of each episode to introduce stochasticity, frame skipping, and frame stacking of the last 4 frames, as is common practice for Atari (Mnih et al. (2015)). For image observations, we use the shared three-layer CNN followed by a fully-connected layer and one-layer fully-connected heads for the policy and value, as described in (Mnih et al. (2015)). For the CNN, we use a Masksembles layer after the first convolutional layer and for the final fully-connected layer before the output action distribution. Dropout/Dropconnect layers are added in eequivalent positions inside the networks. We use 4 sub-models for both Masksembles and Enesembles across experiments. Equivalently, for we use 4 individual samples for uncertainty estimation with MC Dropout/MC Dropconnect. As reported in Appendix-D, we have seen a decrease in reward with increasing dropout probability, with only marginal gains in ROC-AUC Score. We ave therefore decided to keep the dropout probability at 0.2 for all environments.

**Training Details**  During training, we use parallel environments and roll out each sub-model policy at a distinct set of environments, e.g., using sub-model 1 policy for action selection in environments 1 and 2, sub-model 2 as the policy for environments 3 and 4, etc. In the case of Masksembles, the sub-models share a significant part of the network, e.g., the intermediate convolutional layers. Furthermore, the policy and value losses are computed on a shared replay buffer, with transitions sampled equally from rollouts of all sub-models. This common loss function is then applied to optimize all four sub-policies. This is equivalent to the cases of the baseline and Dropout policies and, therefore, requires minor modifications to the existing architectures. For Ensemble networks, $k$ completely independent sub-networks are used. We find that training the ensemble modles completely independently can lead to subpar results. Without any synchronization, both inference and uncertainty estimation become very difficult (for details, see Appendix C). We, therefore, decided to also use a shared replay buffer for ensembles to ensure interoperability of sub-models.

We do not utilize uncertainty estimates during training, although related work has shown that uncertainty can be used to improve training, e.g., by encouraging exploration (Wu et al. (2021); Chen et al. (2017); Osband et al. (2016); Imagawa et al. (2019)). In this paper, we specifically focus on uncertainty estimation during inference.

## A.2  BEST HYPERPARAMETERS

We followed the hyperparameter settings of the *StableBaselines 3-Zoo* (Raffin et al. (2021)) library. These hyperparameter settings were tuned for the Baseline architectures, i.e., they favor the baselines over the adapted architectures. For the MuJoCo environments *HalfCheetah-v3* and *Walker2d-v3*, we used larger layer sizes. We utilized 4 masks for Masksembles, 4 individual models for ensembles, or 4 samples in MC Dropout for each environment.

As mentioned above (in Sec. 4), we followed these settings to validate the simple applicability of the different methods.

In the case of Atari, *layer sharing* means that, e.g., for Masksembles, we apply the masks to the shared feature extractor (CNN) and, therefore, still get 4 predictions for the policy and value outputs. We utilize 4 separate feature extractors with individual policy and value heads for the ensemble.

| Hyperparameters: HalfCheetah-v3 | |
|---|---|
| Algorithm | PPO with GAE |
| Network Arch. | 3-layer MLP, hidden size 256 (384 for Masksembles) |
| Layer Sharing | No layer sharing, Separate policy and value Networks |
| Training Timesteps | 1e6 |
| Num. Parallel Environments | 8 |
| Batch Size | 64 |
| Learning Rate | 2.0633e-5 |
| Num. Epochs | 20 |
| Sampled Steps per Env. | 512 |
| GAE lambda | 0.92 |
| Gamma | 0.98 |
| Ent. Coeff | 0.0004 |
| VF Coeff. | 0.581 |
| Max. Grad Norm | 0.8 |
| Clip Range | 0.1 |
| Obs. Normalization | false |

Table 3: Training hyperparameters for *HalfCheetah-v3*. We chose the larger network size for Masksembles but not for the other methods as we did not see increased performance for the baseline, Dropout, and Ensembles policies.

| Hyperparameters: Ant-v3 | |
|---|---|
| Algorithm | PPO with GAE |
| Network Arch. | 3-layer MLP, hidden size 64 |
| Layer Sharing | No layer sharing, Separate policy and value Networks |
| Training Timesteps | 1e7 |
| Num. Parallel Environments | 8 |
| Batch Size | 32 |
| Learning Rate | 1.90e-5 |
| Num. Epochs | 10 |
| Sampled Steps per Env. | 512 |
| GAE lambda | 0.8 |
| Gamma | 0.98 |
| Ent. Coeff | 4e-7 |
| VF Coeff. | 0.677 |
| Max. Grad Norm | 0.6 |
| Clip Range | 0.1 |
| Obs. Normalization | false |

Table 4: Training hyperparameters for *Ant-v3*

| Hyperparameters: Walker2d-v3 | |
|---|---|
| Algorithm | PPO with GAE |
| Network Arch. | 3-layer MLP, hidden size 128 |
| Layer Sharing | No layer sharing, Separate policy and value Networks |
| Training Timesteps | 1e6 |
| Num. Parallel Environments | 8 |
| Batch Size | 32 |
| Learning Rate | 5.05-5 |
| Num. Epochs | 10 |
| Sampled Steps per Env. | 512 |
| GAE lambda | 0.95 |
| Gamma | 0.98 |
| Ent. Coeff | 0.00059 |
| VF Coeff. | 0.872 |
| Max. Grad Norm | 1 |
| Clip Range | 0.1 |
| Obs. Normalization | true |

Table 5: Training hyperparameters for *Walker2d-v3*. We decided on hidden size 128 as it slightly boosted performance Masksembles and MC Dropout while not substantially impacting baseline performance.

| Hyperparameters: Swimmer-v3 | |
|---|---|
| Algorithm | PPO with GAE |
| Network Arch. | 2-layer MLP, hidden size 256 |
| Layer Sharing | No layer sharing, Separate policy and value Networks |
| Training Timesteps | 1e6 |
| Num. Parallel Environments | 8 |
| Batch Size | 32 |
| Learning Rate | 5.49717e-05 |
| Num. Epochs | 10 |
| Sampled Steps per Env. | 512 |
| GAE lambda | 0.95 |
| Gamma | 0.9999 |
| Ent. Coeff | 0.0554757 |
| VF Coeff. | 0.38782 |
| Max. Grad Norm | 0.6 |
| Clip Range | 0.3 |
| Obs. Normalization | false |

Table 6: Training hyperparameters for *Swimmer-v3*.

| Hyperparameters: Seaquest-v4/MsPacman-v4/Berzerk-v4/SpaceInvaders-v4 | |
|---|---|
| Algorithm | PPO with GAE |
| Network Arch. | 3-layer CNN, fin. hidden size 512 |
| Layer Sharing | Shared CNN, Separate policy and value head layers |
| Training Timesteps | 1e6 |
| Num. Parallel Environments | 8 |
| Batch Size | 256 |
| Learning Rate | 0.00025 (for SpaceInvaders-v4 linear decay) |
| Num. Epochs | 4 |
| Sampled Steps per Env. | 128 |
| GAE lambda | 0.95 |
| Gamma | 0.98 |
| Ent. Coeff | 0.01 (0.2 for Seaquest) |
| VF Coeff. | 0.5 |
| Max. Grad Norm | 1 |
| Clip Range | 0.2 |
| Env. Details | Basis: NoFrameskip-v4, AtariWrapper with frameskip 4, Stacking last 4 frames, Image normalization |

Table 7: Training hyperparameters for *Atari* environments.

## B   Results of the Bi-Objective Meta Optimization

Fig. 4 shows the results of the hyperparameter optimization process. The discrepancy between the ROC-AUC scores reported in Fig. 4 and Tab. 2 can be explained by the different OOD generation processes during hyperparameter optimization and benchmarking of fully trained models. The observation space for the noise-based approach appears to be simpler, i.e., it produces more obvious OOD states compared to the physics-/level-based generation. However, in all cases we observe an advantage of Masksembles over the other approaches.

## C   Results for Ensembles

As the results for Ensembles might be surprising, especially given previous experimental evidence, we have decided to dedicate a detailed discussion in order to explain these findings.

### C.1   Synchronization of Ensembles

We experimented with an independent ensemble model similar to supervised settings for the method comparison. We found that a fully unsynchronized ensemble, i.e., without a shared replay buffer or action output layer, produces vastly different policy and value outputs (see the following Appendix C.2). This can prevent correct (1) averaging across action outputs during inference and (2) comparison of value predictions for uncertainty estimation. Policies can diverge significantly due to randomness during training in the on-policy case, which explains the mismatch between predictions for non-synchronized ensembles.

To resolve this issue, we opted to use a shared rollout buffer. Each individual submodel performs predictions on a distinct set of parallel environments, which is equivalent to the other ensemble-like methods (especially Masksembles). Individual experiences of the sub-models are then collected and used for common optimization. The loss is computed on the shared rollout buffer. This leads to a synchronization effect between models while also ensuring that each sub-model is trained with the complete set of environment interactions.

An explanation of the shared buffer is depicted in Fig. 5.

### C.2   Fully Independent Ensembles Training

In the main part, Appendix F and Appendix C.1, we considered an ensemble implementation with a synchronized shared buffer. In this section, we consider an alternative ensemble implementation: here, all sub-models are trained independently with different seeds and then combined into a common ensemble at the inference phase. This is comparable to supervised ensembles. The main difference is that in the supervised case, all sub-models are trained on the same dataset, whereas, in on-policy RL, the distribution of states an agent is trained with depends on the policy itself. It is, therefore, individual to each sub-model. While significant overlap can occur, it is not guaranteed. In such a case, the sub-models might have learned significantly different policies and value functions. Combining these sub-models into a single ensemble can lead to unreliable predictions: both in terms of action distribution and uncertainty estimation.

**Training Details**   For this independent ensemble training, we utilize the baseline architecture presented in Sec. 4.2 and Fig. 2a. Each of the 4 sub-models is trained with 1/4 of the baseline environment interactions compared to the other methods. For the *MuJoCo* environments *HalfCheetah-v3* and *Swimmer-v3*, each sub-model is trained for 250000 steps, while for *MsPacman-v4* and *SpaceInvaders-4* we trained for 4 million/3 million steps respectively. This ensures that each method has the same amount of environment interactions. To compensate for this, we increased the number of optimization epochs on the training buffer by a factor of 4, meaning that the number of gradient updates for each ensemble sub-model is equivalent to the number of updates for the full training of the other methods. We report average episode reward during shared inference (Tab. 8) and OOD detection performance (Tab. 9). In order to create the Ensemble models, we trained five sub-models and then used each combination of four models as an ensemble model, i.e. $\Pi_1 = \{\pi^1, \pi^2, \pi^3, \pi^4\}, \Pi_2 = \{\pi^1, \pi^2, \pi^3, \pi^5\}$ etc. Instead of three random seeds, we report the reward performance averaged over the 5 possible combinations of sub-models without repetitions.

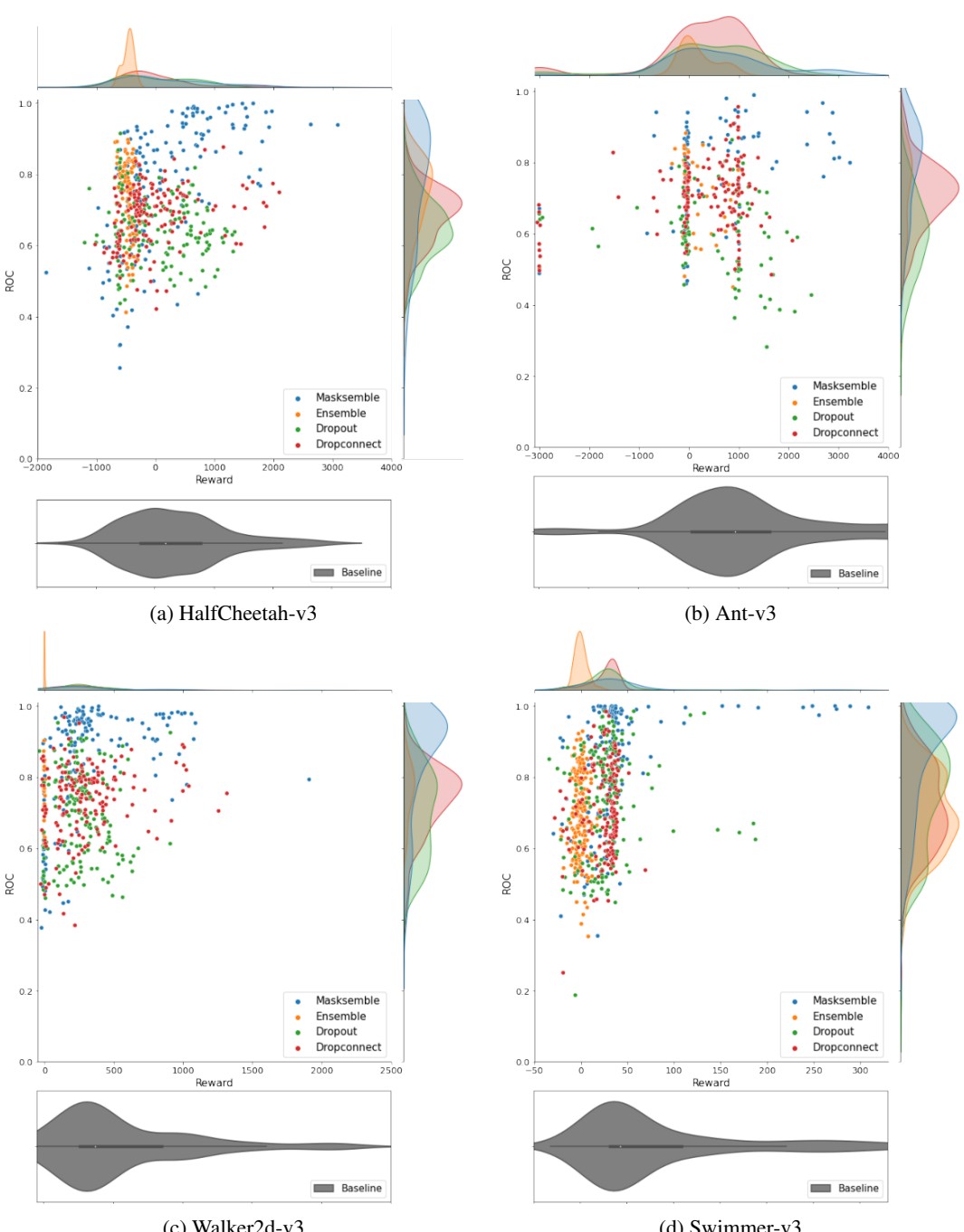

Figure 4: Results of the Hyperparameter Optimization Process. Each dot represents a single run with a corresponding hyper-parameter configuration (e.g. parameters like number of masks/dropout probability, learning rate, model size etc.). The configurations are sampled from a wide space of possible configurations.

**Results**  As can be expected, due to the lower environment interaction budget for sub-models, the overall performance is lower than for comparable methods, although higher than the reported (synchronized) ensemble baseline for the Atari games. For *Swimmer-v3*, the phenomenon of missing synchronization is especially visible: Although the sub-models achieve decent performance on average, the averaged probability distribution produces incoherent behavior, which in turn leads to a significantly lower achieved reward. In terms of OOD detection, we see a drop in performance,

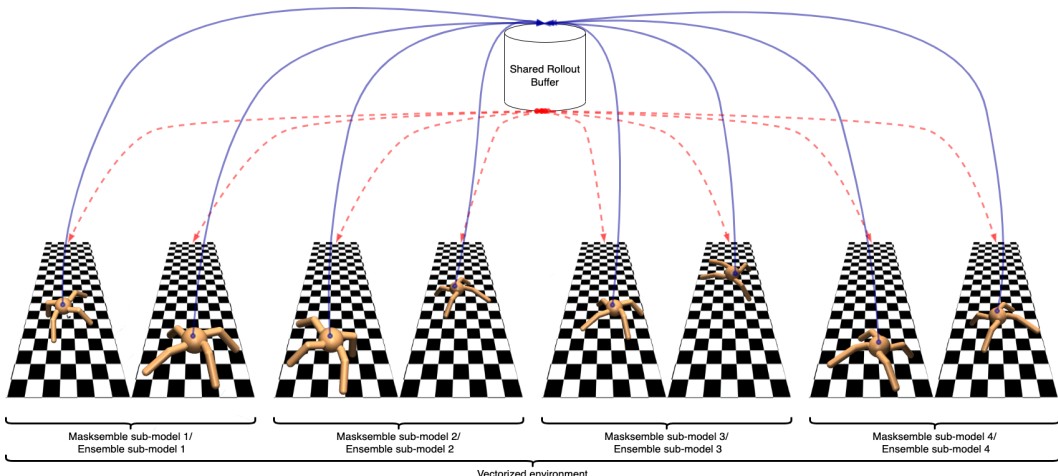

Figure 5: The concept of experience synchronization via a shared rollout buffer. The vectorized environment is divided into groups so that the predicted actions of one Masksembles / ensemble sub-model are executed within each group. The blue arrows mean that all of the agents write their experience to the buffer irrespective of the group they belong to. The red dotted arrows indicate that losses (policy loss/value loss) are computed across all samples in the shared buffer. The shared losses are used to optimize all sub-models.

|  | Walker2d-v3 | Swimmer-v3 | Mspacman-v4 | SpaceInvaders-v4 |
|---|---|---|---|---|
| Independent Ensemble | 435.59 | 0.38 | 1089.2 | 402.3 |
| Independent Ensemble Single | 445.02 | 170.64 | 1473.6 | 434.1 |

Table 8: Average Reward Performance of the independent Ensemble. The average performance can lie significantly below the best single model performance.

especially for the value uncertainty measure. These observations lead us to the architectural choices made in the main paper.

|  | Walker2d-v3 | Swimmer-v3 | Mspacman-v4 | SpaceInvaders-v4 |
|---|---|---|---|---|
| Ind.Ens. Value Uncertainty | 0.34 | 0.26 | 0.161941 | 0.33 |
| Ind.Ens. Policy Uncertainty | 0.64 | 0.39 | 0.67 | 0.36 |
| Ind.Ens. Max. Prob. Uncertainty | 0.40 | 0.28 | 0.53 | 0.51 |

Table 9: OOD Detection ROC scores for the independent ensemble. We notice a significant drop in OOD detection performance, even compared to the ensemble implementation achieving a lower reward.

# D ADDITIONAL RESULTS: ABLATIONS & ADDITIONAL ON-POLICY ALGORITHMS

In this appendix, we want to provide a more in-depth analysis of the hyperparameter sweep results. In particular, we investigate the impact of key design dimensions on reward and OOD detection performance. Secondly, we give results for two additional algorithms: TRPO and A2C.

## D.1 ABLATIONS OF INTER-MODEL CORRELATION/INDEPENDENCE

We want to give some additional key insights into the most important hyperparameter settings.

In the context of our experiments, one of the key aspects we investigated is the effect of the *number of sub-models* for Masksembles and Ensembles, as well as the *dropout probability* for Dropout and Dropconnect. Fig. 9 shows the reward for submodel-based methods. Depending on the environment,

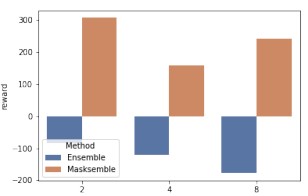
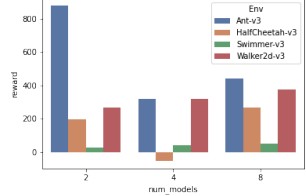
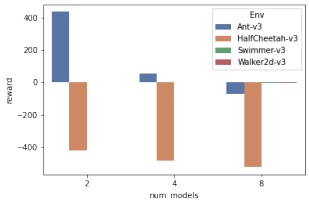

(a) Avg. cumulative episode reward across all runs and environments

(b) Masksembles: Avg. cumulative episode reward split by environment

(c) Ensembles: Avg. cumulative episode reward split by environment

Figure 6: Reward Performance for submodel-based methods with varying **number of models**: We see an stagnating or slightly negative trend for adding additional sub-models.

we might see an advantage for methods with a lower number of models. As hypothesized above, one reason might be that a lower number of models might be more on-policy, i.e., a larger share of steps is contributed by each individual submodel. For Masksembles, a secondary effect is that the submodel size shrinks with additional masks, as the total number of units in a layer stays constant, which could affect performance as well. For ROC-AUC, we see a clear improvement when adding additional models, i.e., increasing the number of models improves the quality of OOD detection.

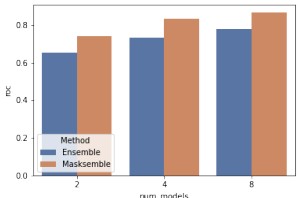
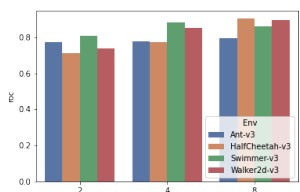
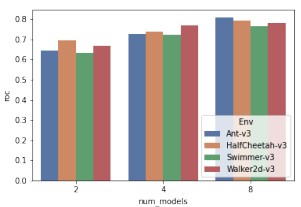

(a) Avg. ROC-AUC scores across all runs and environments

(b) Masksembles: Avg. ROC-AUC split by environment

(c) Ensembles: Avg. ROC-AUC split by environment

Figure 7: ROC-AUC for submodel-based methods with varying **number of models**. Here we observe a clear positive effect of adding additional models on the quality of uncertainty estimation.

A partially equivalent parameter for Dropout and Dropconnect is the *dropout probability* $p$, i.e., the probability of deactivating a neuron/weight during a single inference run. A higher $p$ corresponds to less correlated/individual models. Other than for the submodel-based methods, we do not see a noticeable improvement in OOD detection performance when increasing the *dropout probability*.

## D.2 OTHER ALGORITHMS

We chose PPO as a representative on-policy actor-critic RL algorithm due to its widespread use across research projects. However, we think that the presented results are representative for other on-policy

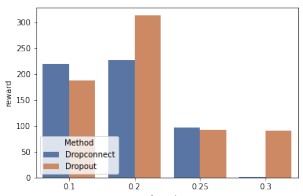
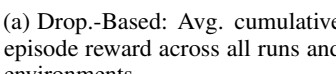

(a) Drop.-Based: Avg. cumulative episode reward across all runs and environments

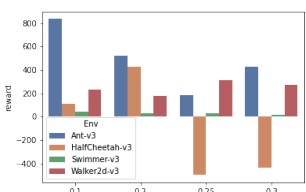

(b) Dropout: Avg. cumulative episode reward split by environment

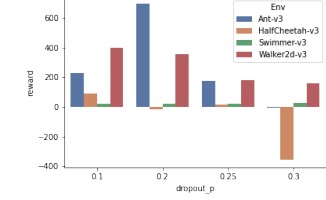

(c) Dropconnect: Avg. cumulative episode reward split by environment

Figure 8: Reward Performance for submodel-based methods with varying **Dropout Prob.**: $p = 0.2$ can perform well, but a lower $p$ is generally preferable. Still, the reward is noticeably below baseline, which corresponds to $p = 0.0$.



(a) Avg. ROC-AUC scores across all runs and environments

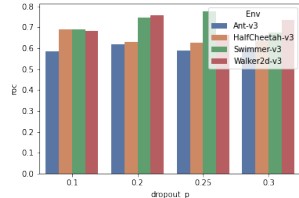

(b) Dropout: Avg. ROC-AUC split by environment

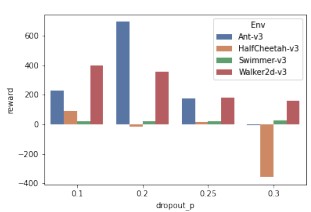

(c) Dropconnect: Avg. ROC-AUC split by environment

Figure 9: ROC-AUC for dropout-based methods with varying **Dropout Prob.**.

RL methods. In the following, we present results of the presented hyperparameter-optimization procedure for TRPO (Schulman et al. (2015)) and A2C (Mnih et al. (2016); Sutton & Barto (2018)). As we can see if Fig. 10, Masksembles still outperforms the other approaches but has a significant variance. Furthermore, we don't see a clear advantage in terms of OOD detection performance, although some runs still perform very well. Additionally, all sample-based methods seem to lack behind the baseline. Although the results for TRPO and A2C are weaker compared to the PPO, they still confirm the general trend. However, we assume that both algorithms are even more impacted by the "off-policy-ness" of the sub-model/dropout-based approaches. Especially A2C produces very weak results with high learning rates and gradient clipping ranges, which can cause a complete failure of the training. However, while Masksemble produces some of the weakest results, it also achieves the overall best training results, showing again the potential of Masksembles to be optimized to a strong configuration. As A2C is impacted by this in particular, we hypothesize that the (approximated) trust region optimization of TRPO and PPO can mitigate some of the worst effects by stabilizing the individual policies of the sub-models. Investigating the particularities of other algorithms and their interplay with the presented sample-based methods is a potential future line of work.

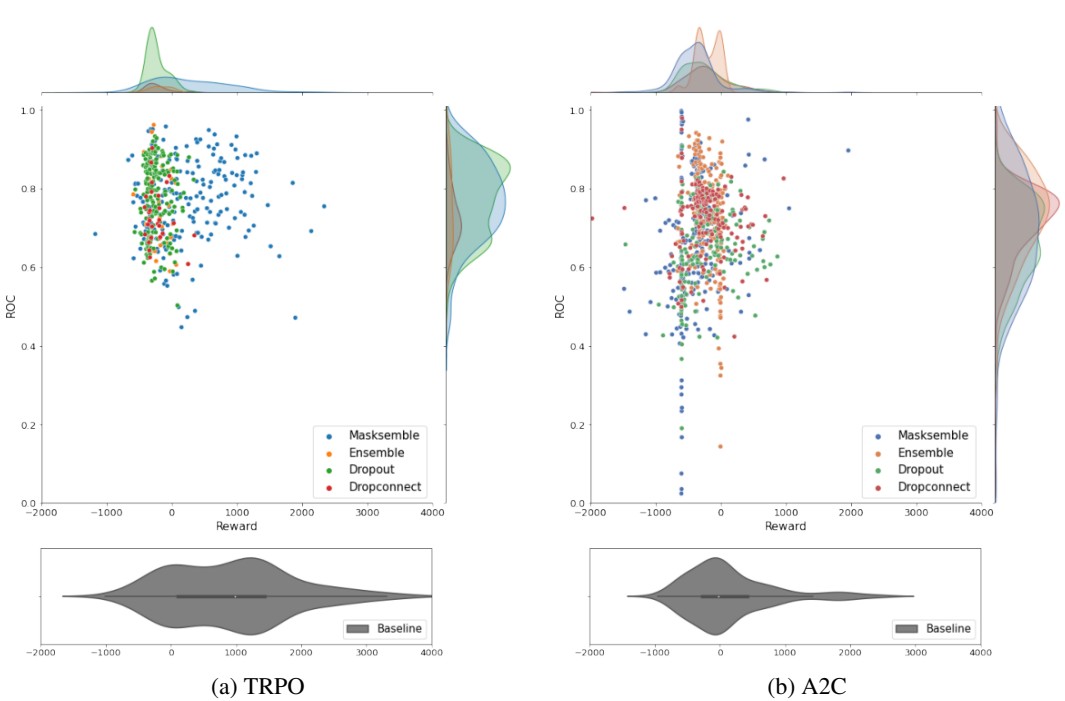

(a) TRPO

(b) A2C

Figure 10: Results of the Hyperparameter Optimization Process for the *Half-Cheetah-v3* environment for TRPO and A2C.

# E   ALTERNATIVE POLICY UNCERTAINTY MEASURES BASED ON JENSEN-SHANNON DIVERGENCE

In Sec. 3.2, we use the standard deviation of predicted means of actions of the policy distributions in the continuous case; as well as the standard deviation of action probabilities of the policy distributions in the discrete case, as policy uncertainty measure.

Here we investigate Jensen-Shannon divergence between output distributions as an alternative measure for policy uncertainty, defined as follows for categorical (Eq. 7) and continuous action distributions (Eq. 8):

$$\mathcal{U}_{cat,JS}^{\Pi}(s) = \max_{i,j \in 1,..,k} JS(\pi^i(a|s)||\pi^j(a|s)) =$$

$$= \max_{i,j \in 1,..,k} \left\{ \frac{1}{2} \left[ KL(\pi^i(a|s)||\pi^j(a|s)) + KL(\pi^j(a|s)||\pi^i(a|s)) \right] \right\} \quad (7)$$

and

$$\mathcal{U}_{con,JS}^{\Pi}(s) = \max_{i,j \in 1,..,k} JS(\pi^i(a|s)||\pi^j(a|s)) = \max_{i,j \in 1,..,k} \left\{ \frac{1}{2} \left[ ||\tilde{\pi}^i(a|s) - \tilde{\pi}^j(a|s)||_2 \right] \right\} \quad (8)$$

We compute the pairwise JS-divergence between the $k$ categorical probability distributions with $N$ actions for the discrete case. For continuous actions, we only compare the distance of mean vectors because it is proportional to the JS divergence of the action distributions.

To show that we utilize the fact that the covariance matrices are *predefined*, *scalar* and *equal*.

Let $P = \mathcal{N}_d(\mu_1, \Sigma_1)$ and $Q = \mathcal{N}_d(\mu_2, \Sigma_2)$, and calculate Kullback-Leibler divergence between two multivariate normal distributions in a stragthforward way:

$$KL(P||Q) = \int_x P(x) \log \frac{P(x)}{Q(x)} dx =$$

$$= \int_x P(x) \log \left[ \frac{1}{(2\pi)^{d/2}|\Sigma_1|^{1/2}} \exp\left( -\frac{1}{2}(x - \mu_1)^T \Sigma_1^{-1} (x - \mu_1) \right) \right] -$$

$$- \log \left[ \frac{1}{(2\pi)^{d/2}|\Sigma_2|^{1/2}} \exp\left( -\frac{1}{2}(x - \mu_2)^T \Sigma_2^{-1} (x - \mu_2) \right) \right] dx =$$

$$= \frac{1}{2} \int_x P(x) \left[ \log \frac{|\Sigma_2|}{|\Sigma_1|} - (x - \mu_1)^T \Sigma_2^{-1} (x - \mu_1) + (x - \mu_2)^T \Sigma_2^{-1} (x - \mu_2) \right] dx =$$

$$= \frac{1}{2} \left[ \mathbb{E}_{P(x)} \log \frac{|\Sigma_2|}{|\Sigma_1|} - \text{tr} \left\{ \mathbb{E}_{P(x)} \left[ (x - \mu_1)(x - \mu_1)^T \right] \Sigma_1^{-1} \right\} + \mathbb{E}_{P(x)} \left[ (x - \mu_2)^T \Sigma_2^{-1} (x - \mu_2) \right] \right] =$$

$$\overset{\substack{tr\mathbb{E}_P(c)=\mathbb{E}_P tr(c), \\ tr(ABC)=tr(BCA) \\ \Sigma_1=\left[(x-\mu_1)(x-\mu_1)^T\right]}}{=} \frac{1}{2} \left[ \log \frac{|\Sigma_2|}{|\Sigma_1|} - \text{tr}\left\{ \Sigma_1 \Sigma_1^{-1} \right\} + \mathbb{E}_{P(x)} \left[ (x - \mu_2)^T \Sigma_2^{-1} (x - \mu_2) \right] \right] =$$

$$\overset{tr(\mathbb{I}_d)=d}{=} \frac{1}{2} \left[ \log \frac{|\Sigma_2|}{|\Sigma_1|} - d + \text{tr}\left\{ \Sigma_2^{-1}\Sigma_1 \right\} + (\mu_2 - \mu_1)^T \Sigma_2^{-1} (\mu_2 - \mu_1) \right] \quad (9)$$

in the last transition, we used the fact:

$$\mathbb{E}_{P(x)} \left[ (x - \mu_2)^T \Sigma_2^{-1} (x - \mu_2) \right] = \text{tr}\left\{ \Sigma_2^{-1}\Sigma_1 \right\} + (\mu_2 - \mu_1)^T \Sigma_2^{-1} (\mu_2 - \mu_1)). \quad (10)$$

We can use result from Eq. 9 to calculate Jensen-Shannon divergence:

$$JS(P||Q) = \frac{1}{2} \left\{ KL(P||Q) + KL(Q||P) \right\} =$$

$$= \frac{1}{2} \{ \frac{1}{2} [log \frac{|\Sigma_1|}{|\Sigma_2|} - d + tr(\Sigma_2^{-1}\Sigma_1) + (\mu_2 - \mu_1)^T \Sigma_2^{-1}(\mu_2 - \mu_1)] +$$

$$+ \frac{1}{2} [log \frac{|\Sigma_2|}{|\Sigma_1|} - d + tr(\Sigma_1^{-1}\Sigma_2) + (\mu_1 - \mu_2)^T \Sigma_1^{-1}(\mu_1 - \mu_2)] \}. \quad (11)$$

Notice, that in our case: $\Sigma_1 = \Sigma_2 = \sigma\mathbb{I}$. It implies:

$$tr(\Sigma^{-1}\Sigma) = tr(\mathbb{I}) = d \tag{12}$$

and

$$x^T\Sigma x = \sigma||x||_2^2 \ \forall x \in \mathbb{R}^d \tag{13}$$

By substituting Eq. 12 and Eq. 13 to Eq. 11 we get:

$$JS(P||Q) = \frac{1}{2}\left\{\sigma||\mu_2 - \mu_1||_2^2 + \sigma||\mu_1 - \mu_2||_2^2\right\} = \sigma||\mu_1 - \mu_2||_2^2 \tag{14}$$

that concludes our statement.

The results for policy uncertainty for continuous action space corresponded to Eq. 7 are depicted on Fig. 11.

The results for policy uncertainty for discrete action space corresponded to Eq. 8 action space are depicted on Fig. 12.

We find that the alternative uncertainty measure produces very similar estimates to the one provided in Eq. 3 and Eq. 4 (compare to Fig. 23 and Fig. 28). Both policy uncertainty measures gave similar results within a margin of error in all considered environments. We, therefore, decided to present the more conceptually simple measure in the main part.

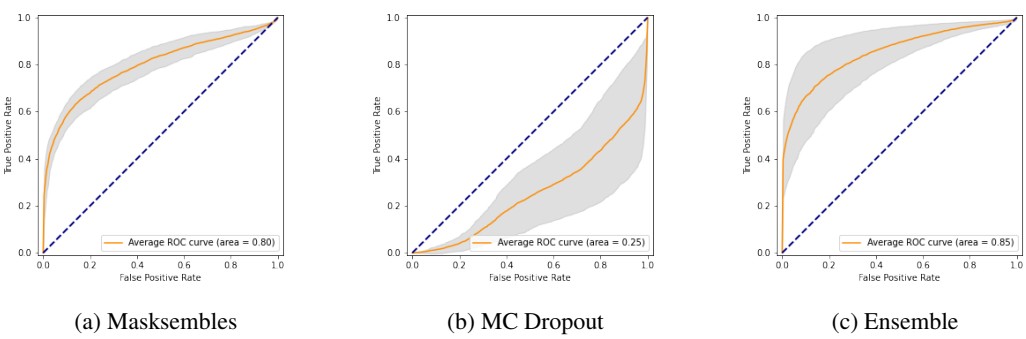

| (a) Masksembles | (b) MC Dropout | (c) Ensemble |

Figure 11: ROC curve for alternative JS-based policy uncertainty for *HalfCheetah-v3* environment

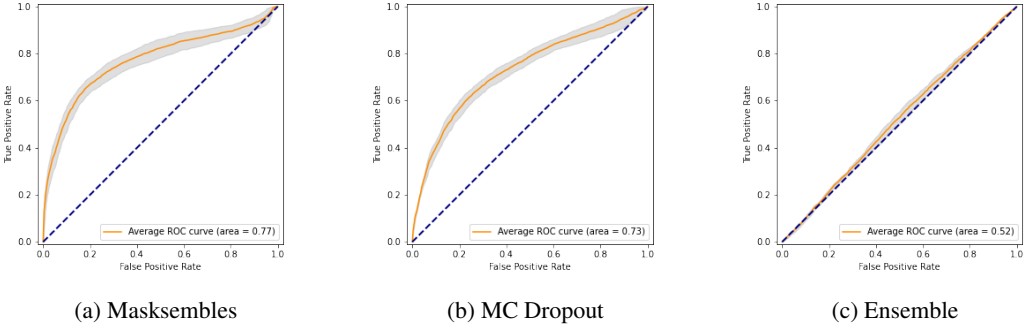

| (a) Masksembles | (b) MC Dropout | (c) Ensemble |

Figure 12: ROC curve for alternative JS-based policy uncertainty for *Seaquest-v4* environment

# F INVESTIGATION OF SCALING ABILITIES OF MASKSEMBLES, MC DROPOUT, AND ENSEMBLES

**Scaling Behavior** When scaling the hidden layer size, we observe that only Masksembles show significantly increased performance. We hypothesize that the scaling can partially compensate for the

smaller layer size due to the introduction of masks. We like to note that we do not observe the same scaling properties for Dropout and Ensemble. For the baseline model, we expect the performance to be either lower or equal compared to the smaller network size, as the model size has been tuned as part of previous hyperparameter tuning. We evaluated the scaling properties in the context of MuJoCo. Results are presented in Fig. 13 and Fig. 14.

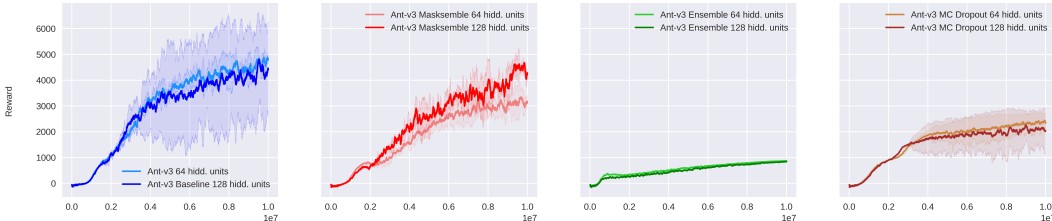

Figure 13: Scaling properties of different methods in *Ant-v3*: We find that only Masksembles show improved performance when increasing the network size. The other methods may even drop in performance when scaling the layers. Results averaged across 3 runs for 64 hidden units and 2 runs for 128 hidden units.

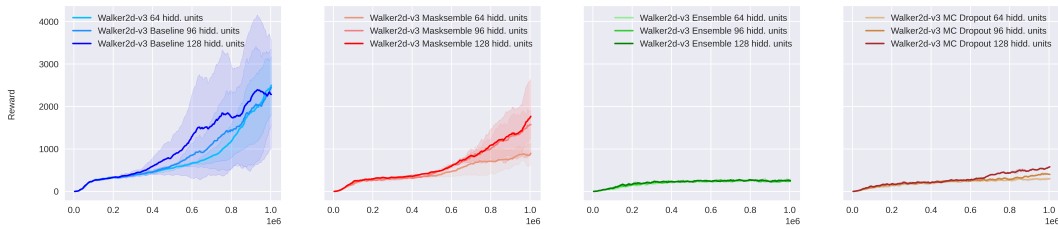

Figure 14: Scaling properties of different methods in *Walker2d-v3*: We evaluate the scaling across 64 (default), 96, and 128 hidden units. We find a clear improvement for Masksembles, and a slight improvement for MC Dropout, while the baseline and Ensemble policies are mainly unaffected. Although the baseline policy achieves a higher reward earlier, the final reward of the baseline policy is very close to the original one.

## G    VALUE UNCERTAINTY AND POLICY UNCERTAINTY ROC CURVES

In addition to ROC curve for value uncertainty for *HalfCheetah-v3* in Fig. 15 we also provide ROC curve for policy uncertainty (please see Fig. 23).

Apart from that, we provide value uncertainty (corresponded to Eq. 6)and policy uncertainty (corresponded to Eq. 3 for continuous case and Eq. 4 for discrete action space) ROC curves for the following environments: Ant-v3 (see Fig. 16, 24), Walker-v3 (see Fig. 17, 25), Swimmer-v3 ( 18, 26), Pacman-v4 (see Fig. 19, 27), Seaquest-v4 (see Fig. 20,28), Berzerk-v4 (see Fig. 21,29), and SpaceInvaders

It is important to note that ROC curves for ensembles could be invalid (see Fig. 16d, 24d) because the qualities of the Ensembles policies are quite poor in general. As a result, it cannot validly differentiate between ID and OOD states.

Another important notice regarding the "convex shape" of ROC curves for MC Dropout (see Fig. 16b, 23b). In the case of simple binary classification, labels are nothing but marking the class belongings, and changing the label "0" to "1" and vice versa will not affect the classification results. This *symmetry* leads to *symmetry* of the ROC curve with respect to the diagonal ROC curve (that is, the ROC curve corresponded to a random binary classifier), and it means that labels should be inverted. After the inversion of labels, the ROC curve will have a normal "concave shape."

A convex curve, in the case of binary classification, for ID and OOD states, actually reveals the inability of the model to correctly identify OOD states, which is particularly dangerous because the model is more confident in its behavior in these states than in known distribution.

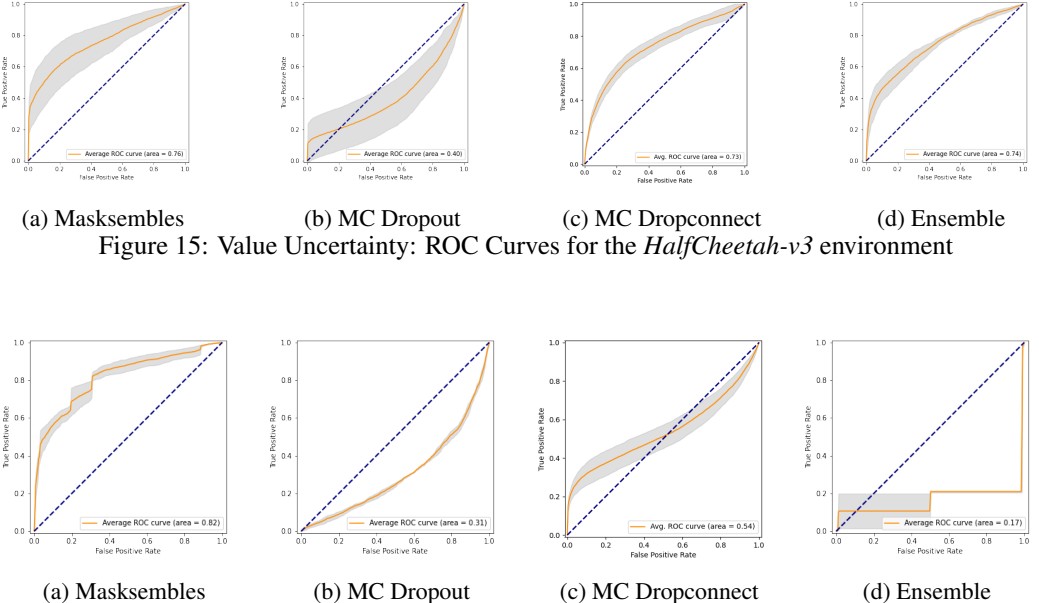

(a) Masksembles      (b) MC Dropout      (c) MC Dropconnect      (d) Ensemble

Figure 15: Value Uncertainty: ROC Curves for the *HalfCheetah-v3* environment

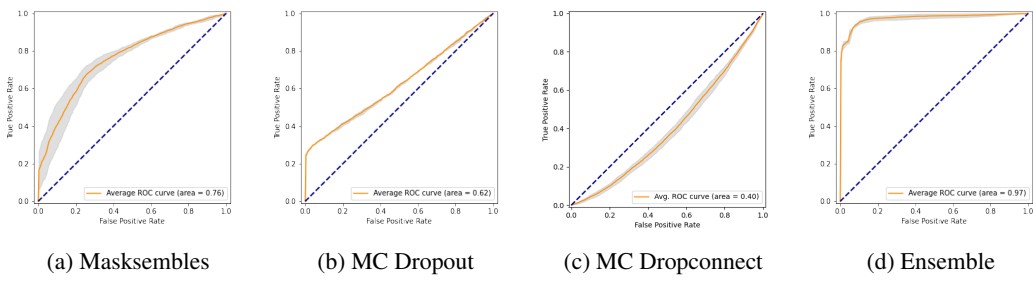

(a) Masksembles      (b) MC Dropout      (c) MC Dropconnect      (d) Ensemble

Figure 16: Value Uncertainty: ROC Curves for the *Ant-v3* environment

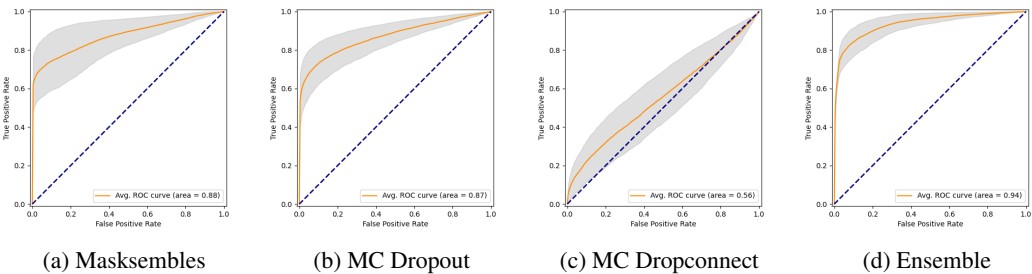

(a) Masksembles      (b) MC Dropout      (c) MC Dropconnect      (d) Ensemble

Figure 17: Value Uncertainty: ROC Curves for the *Walker2d-v3* environment

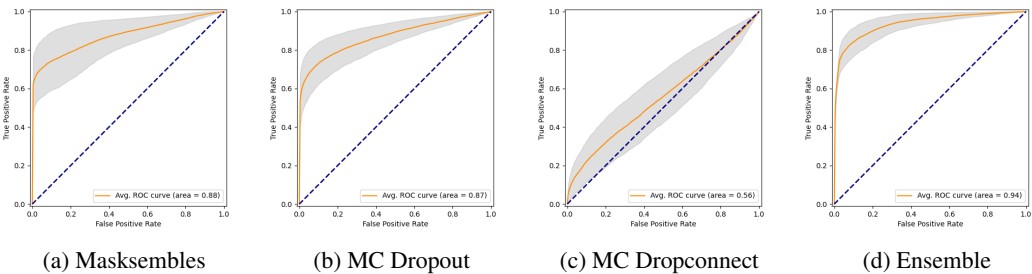

(a) Masksembles      (b) MC Dropout      (c) MC Dropconnect      (d) Ensemble

Figure 18: Value Uncertainty: ROC Curves for the *Swimmer-v3* environment

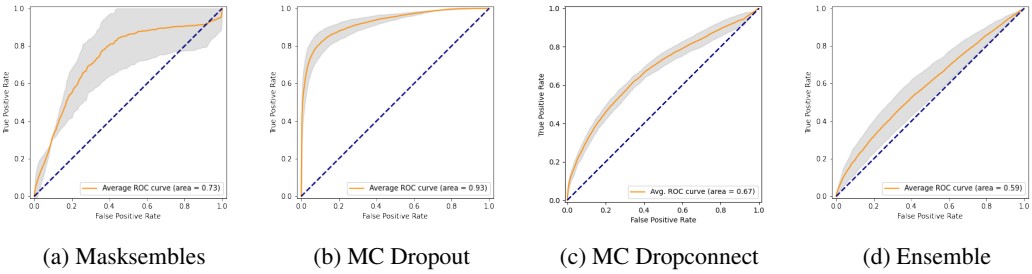

Figure 19: Value Uncertainty: ROC Curves for the *MsPacman-v4* environment

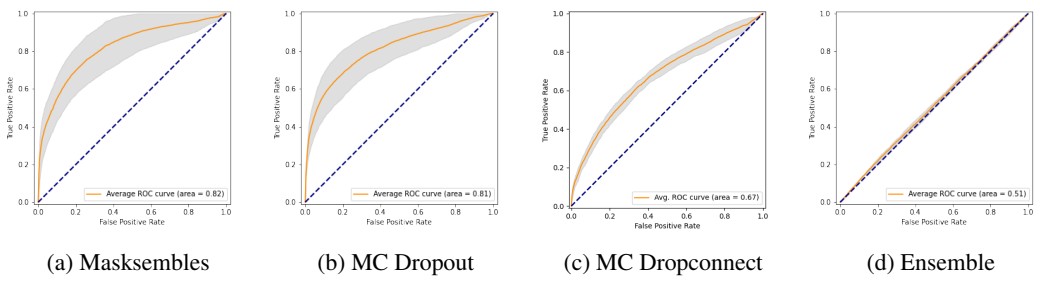

Figure 20: Value Uncertainty: ROC Curves for the *Seaquest-v4* environment

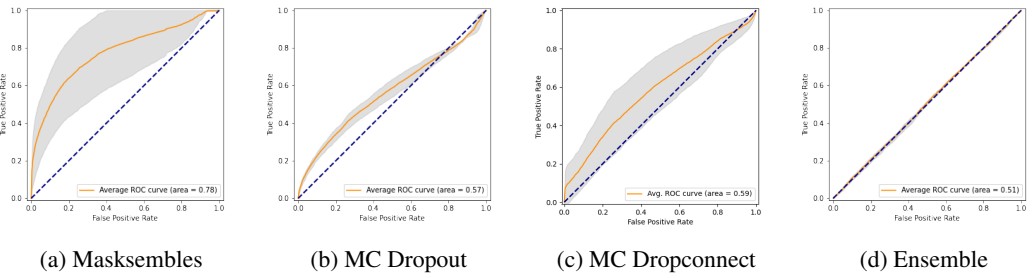

Figure 21: Value Uncertainty: ROC Curves for the *Berzerk-v4* environment

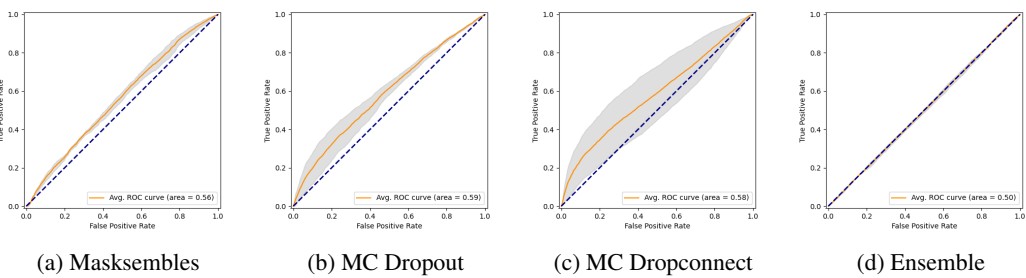

Figure 22: Value Uncertainty: ROC Curves for the *SpaceInvaders-v4* environment

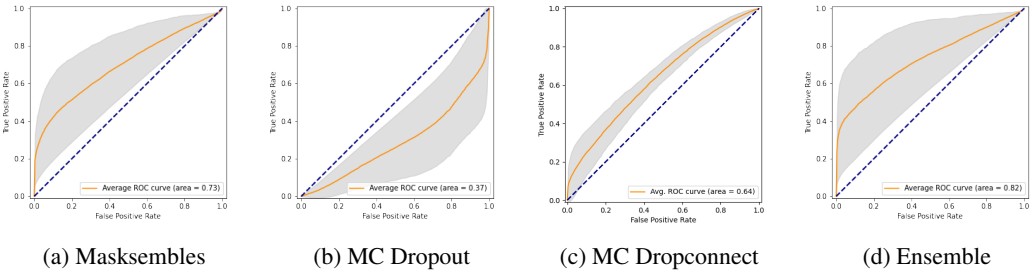

Figure 23: Policy Uncertainty: ROC Curves for the *HalfCheetah-v3* environment

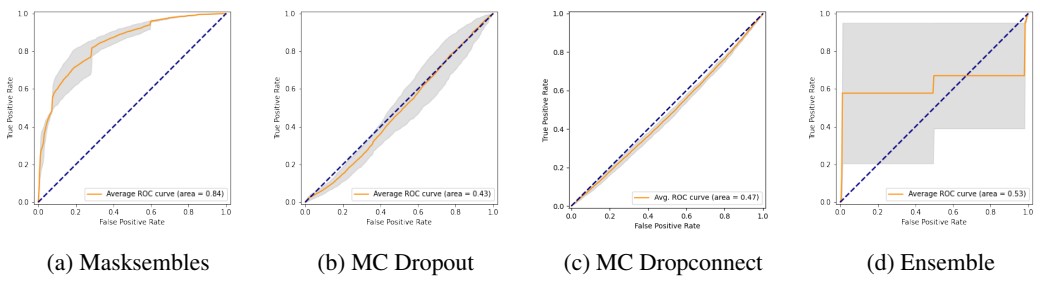

Figure 24: Policy Uncertainty: ROC Curves for the *Ant-v3* environment

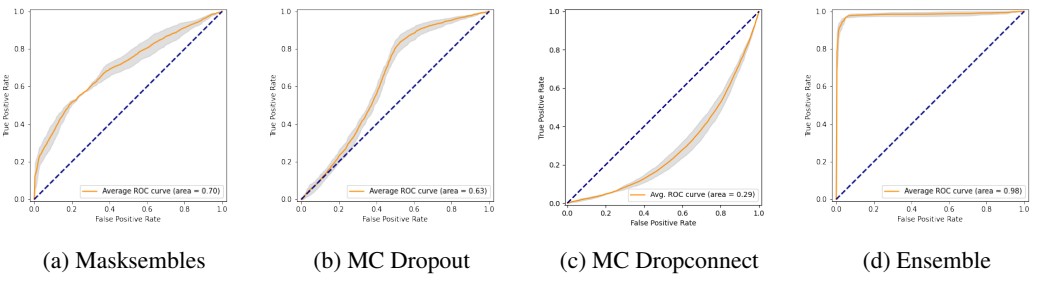

Figure 25: Policy Uncertainty: ROC Curves for the *Walker2d-v3* environment

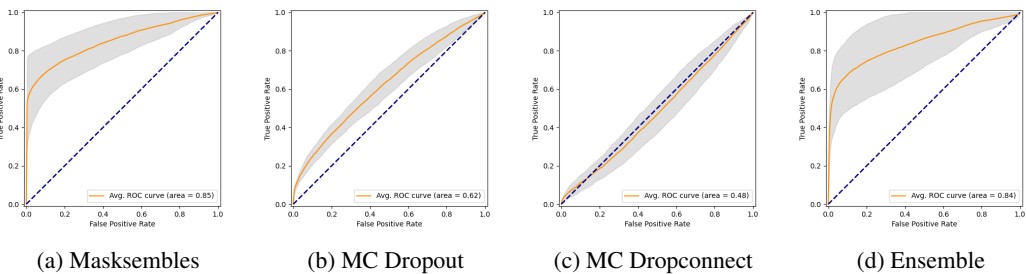

Figure 26: Policy Uncertainty: ROC Curves for the *Swimmer-v3* environment

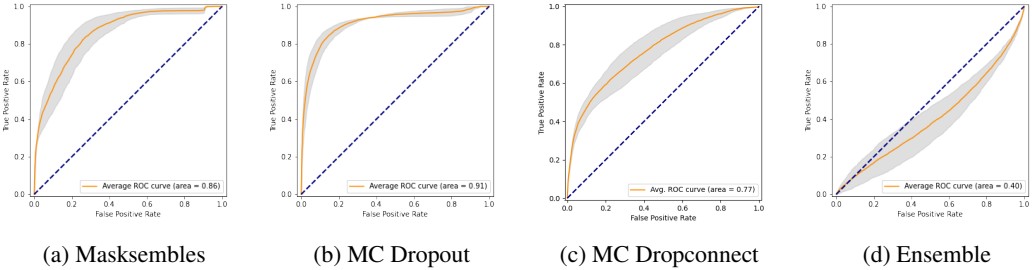

(a) Masksembles     (b) MC Dropout     (c) MC Dropconnect     (d) Ensemble

Figure 27: Policy Uncertainty: ROC Curves for the *MsPacman-v4* environment

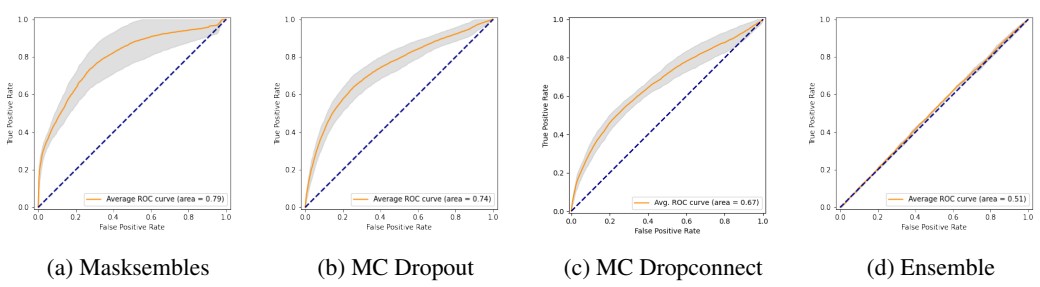

(a) Masksembles     (b) MC Dropout     (c) MC Dropconnect     (d) Ensemble

Figure 28: Policy Uncertainty: ROC Curves for the *Seaquest-v4* environment

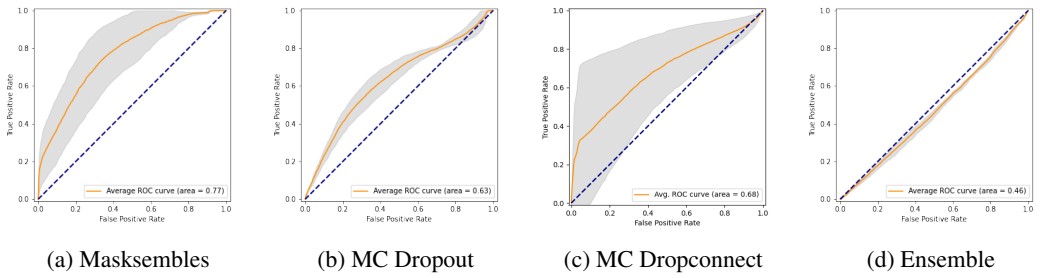

(a) Masksembles     (b) MC Dropout     (c) MC Dropconnect     (d) Ensemble

Figure 29: Policy Uncertainty: ROC Curves for the *Berzerk-v4* environment

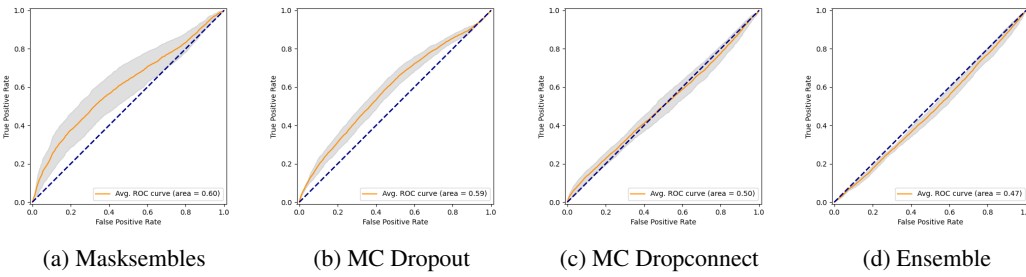

(a) Masksembles     (b) MC Dropout     (c) MC Dropconnect     (d) Ensemble

Figure 30: Policy Uncertainty: ROC Curves for the *SpaceInvaders-v4* environment

## H    EXAMPLES OF OUT-OF-DISTRIBUTION STATES AND PHYSICS INTERVENTION DETAILS

To compute the ROC curves, we sampled 2000 steps from the trained policy in the default environment as in-distribution-states. We then sample ca. 2000 OOD states either from the expert dataset or generated in a physically modified environment.

**MuJoCo**    To create OOD samples for Mujoco environments, we use interventions on physics(See Tab .10). Examples of ID and OOD samples are depicted in Appendix H. We independently sampled 5 configurations, trained and executed the policies in these modified environments for a maximum of 500 steps, after which an episode was restarted with a modified configuration. The first 10 states of each episode were removed from the samples. Because the modified physics partially came into effect only after the agent started performing actions, the perturbed early states might be very similar to the original early states.

**Atari**    For Atari, we sample OOD states from human gameplay, recorded for the Atari HEAD (Zhang et al. (2019)) dataset. Multiple factors contribute to the states being OOD: (1) The human gameplay does generally not fully overlap with agent gameplay, which means the agent will often, e.g., be in screen positions that are not part of the trajectories produced by trained agents, (2) we sample states only from advanced levels that have significant differences to the training distribution, e.g., additional elements or changed layouts. We verify, via qualitative inspection, that the trained agents are not able to achieve high performance in presented OOD states: as an example, the level in *MsPacman-v4* can be selected at initialization, and therefore trained agents can be tested in this case.

| | Gravity (z-direction) (in $\frac{m}{s^2}$) | Wind (in $\frac{m}{s}$) | Friction Coeff. | Geom. Element Scaling |
|---|---|---|---|---|
| HalfCheetah-v3 | $[0.5 - 4]$ | $[0 - 1]$ | $[0.1 - 50]$ | All elements of legs individually scaled with random factors $[1.5 - 2.5]$ |
| Ant-v3 | $[0.5 - 4]$ | $[0 - 1]$ | $[0.1 - 50]$ | Each leg is individually scaled with random factors $[1.5 - 2.5]$ |
| Walker2d-v3 | $[0.5 - 4]$ | $[0 - 1]$ | $[0.1 - 50]$ | Left/right leg is scaled as a whole with random factors $[1.5 - 2.5]$ |
| Swimmer-v3 | $[0.5 - 4]$ (also x-dir) | $[0 - 1]$ | $[0.1 - 50]$ | All three body elements are scaled with random factors $[1.5 - 2.5]$ |

Table 10: We sample random configurations from the presented parameter ranges. A configuration, therefore, encompasses a random number of different modifications to the environment. The wind parameter influences resistance/friction inside the medium.

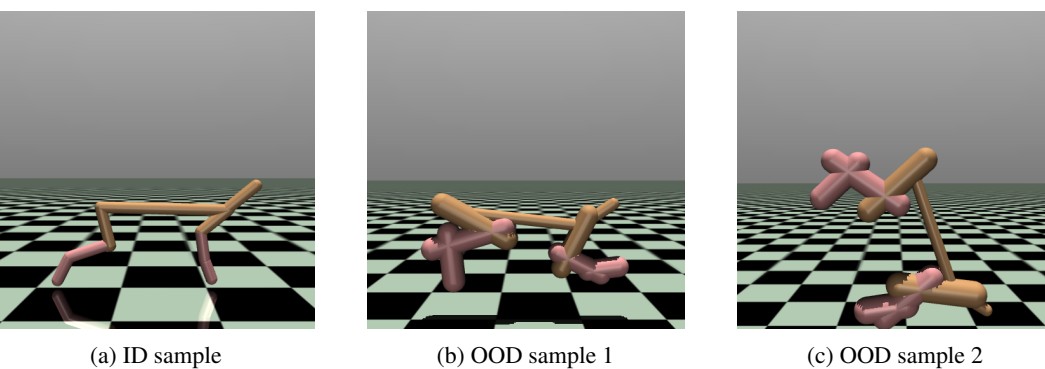

|          (a) ID sample          |          (b) OOD sample 1          |          (c) OOD sample 2          |

Figure 31: ID and OOD states for *MuJoCo: HalfCheetah-v3*. Elements of the body are randomly resized, which leads to vastly different effects. The two OOD samples are recorded at the same timestep within an episode, highlighting the different effects the modifications can have on the execution of the policy.

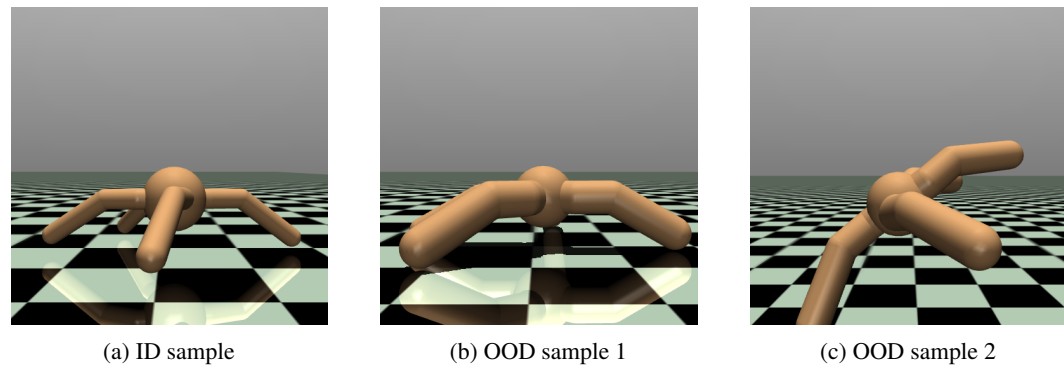

(a) ID sample        (b) OOD sample 1        (c) OOD sample 2

Figure 32: ID and OOD states for *MuJoCo: Ant-v3*. Elements of the body are randomly resized, which leads to vastly different effects.

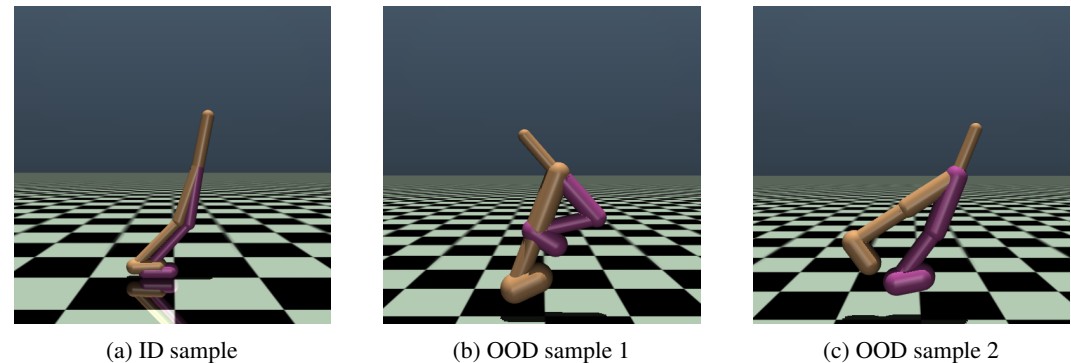

(a) ID sample        (b) OOD sample 1        (c) OOD sample 2

Figure 33: ID and OOD states for *MuJoCo: Walker2d-v3*. Elements of the body are randomly resized, which leads to vastly different effects.

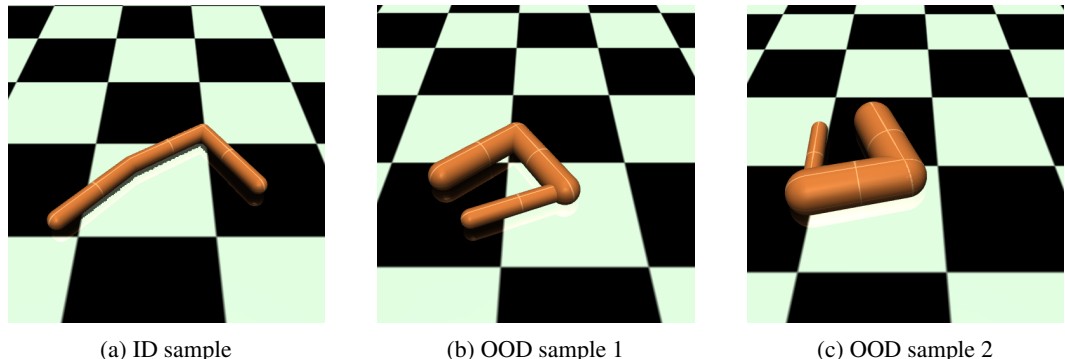

(a) ID sample        (b) OOD sample 1        (c) OOD sample 2

Figure 34: ID and OOD states for *MuJoCo: Swimmer-v3*. Elements of the body are randomly resized, which leads to different observation inputs.

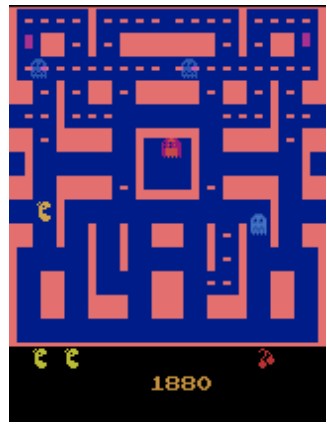
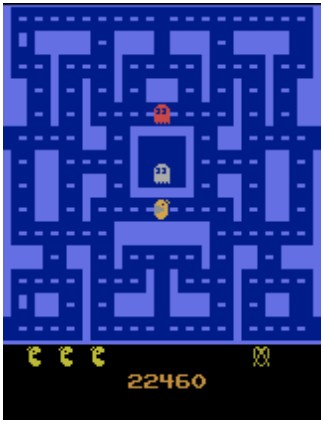
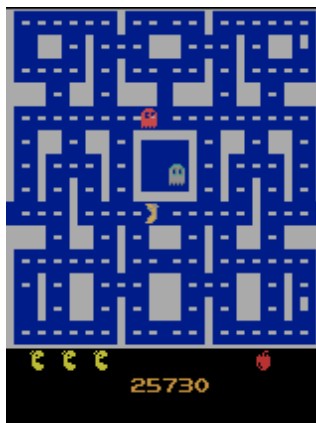

(a) Frame from ID sample | (b) Frame from OOD sample 1 | (c) Frame from OOD sample 2

Figure 35: ID and OOD samples for *Atari: MsPacman-v4*. In later stages, the level layout is fundamentally different. Trained agents only perform very poorly in these new layouts, i.e., we find that they do not generalize to these states and therefore treat them as out-of-distribution. Other details like the enemy ship above the water line or changing the color of fish also occur at later stages of the game.

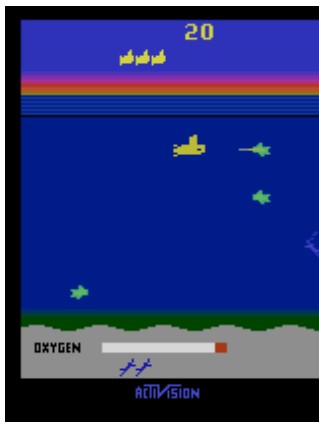
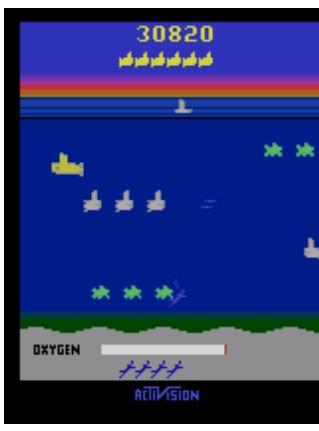
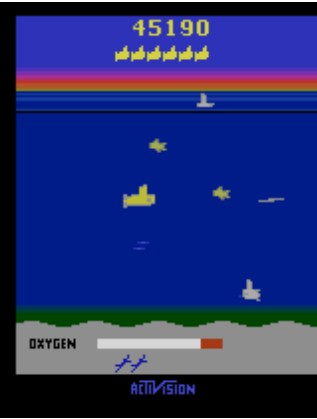

(a) Frame from ID sample | (b) Frame from OOD sample 1 | (c) Frame from OOD sample 2

Figure 36: ID and OOD samples for *Atari: Seaquest-v4*. While core parts are shared (i.e., the core gameplay), important on-screen elements are not visible in ID states, e.g., none of the trained RL agents manages to collect a large number of divers or extra lives (yellow submarines at the top of the screen).

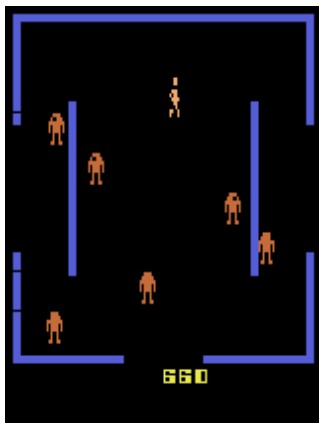 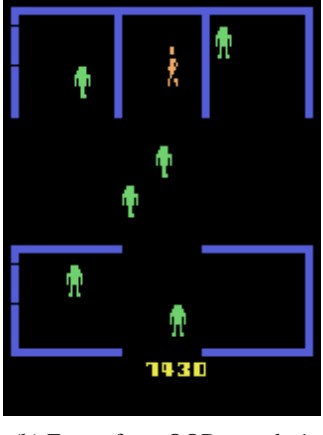 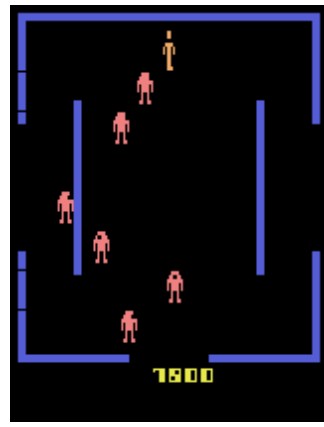

(a) Frame from ID sample     (b) Frame from OOD sample 1     (c) Frame from OOD sample 2

Figure 37: ID and OOD samples for *Atari: Berzerk-v4*. Layouts can be different but can also match in later stages. In the second sample, the positions of agents and enemies, as well as the score, are different from the earlier ID samples.

# I  ADDITIONAL OUT-OF-DISTRIBUTION DETECTION RESULTS

In this section, we provide results regarding additional pixel perturbations. An average uncertainty score for ID states is sampled via rolling out the policy in the environment. This averaged uncertainty score is taken as a baseline for each method (score 1.0). We then compute the uncertainty score for: (1) Predicting based on a white frame, (2) based on a black frame, based on the original with increasing levels of added Gaussian noise: (3) $\pm 50$, (4) $\pm 100$, and (5) $\pm 150$. Lastly, we experimented with applying a custom mask to the input image (e.g., masking out areas of the game frame). As a basis, we chose the *SeaquestNoFrameskip-v4* environment. We report both the effect on value uncertainty (see Fig. 38) and policy uncertainty (see Fig. 39). Compared to the average ID uncertainty score (reported as score 1.0 on a per-method basis), the uncertainty scores are noticeably higher than the baseline score (averaged across non-corrupted ID states). The results on the extreme cases (black/white frame), as well as predicting on states with very high noise levels) show that the predicted uncertainty is not necessarily reliable. We based our decision to move to an *advanced game-state* attack on these initial results, as we found, e.g., that adding Gaussian noise is not a good representation for OOD states.

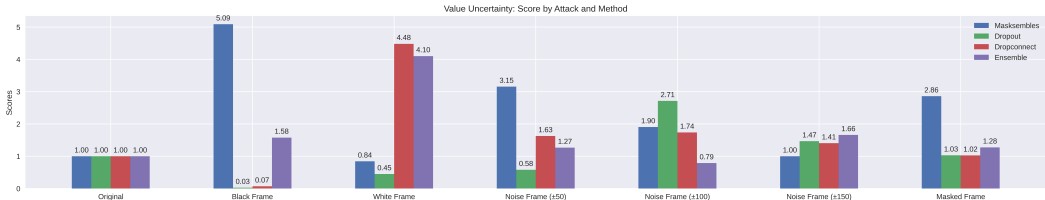

Figure 38: The effect of different pixel perturbations on value uncertainty. *From left-to-right*: no perturbation, replacement to black frame, replacement to white frame, adding Uniform Noise $[-50; 50]$, adding Uniform Noise $[-100; 100]$, adding Uniform Noise $[-150; 150]$, hiding parts of the images with the static mask.

**Uncertainty Measure Curves**  For the benchmark runs, we can look at the temporal structure of the value estimates. For *HalfCheetah-v3* and *Seaquest-v3*, we present exemplary plots that show the magnitudes of the uncertainty measures presented in the main part. In each plot, the first 2000 steps correspond to ID states, whereas the following steps correspond to OOD steps.

In the Fig. 40 we provide policy uncertainty and value uncertainty plots for *HalfCheetah-v3*.

In the Fig. 41 we provide policy uncertainty and value uncertainty plots for *Swimmer-v3*.

Figure 39: The effect of different pixel perturbations on policy uncertainty. *From left-to-right*: no perturbation, replacement to black frame, replacement to white frame, adding Uniform Noise $[-50; 50]$, adding Uniform Noise $[-100; 100]$, adding Uniform Noise $[-150; 150]$, hiding parts of the images with the static mask.

In the Fig. 42 we provide policy uncertainty and value uncertainty plots for *MsPacmanNoFrameskip-v4*.

In the Fig. 43 we provide policy uncertainty and value uncertainty plots for *SeaquestNoFrameskip-v4*.

We observe that Masksembles are better for detecting OOD samples compared to MC Dropout and Ensemble. We see that the uncertainty level for OOD samples is much higher than for ID in the case of Masksembles.

Apart from that, we observe that value uncertainty and policy uncertainty are much better synchronized compared to Dropout, Dropconnect and Ensemble.

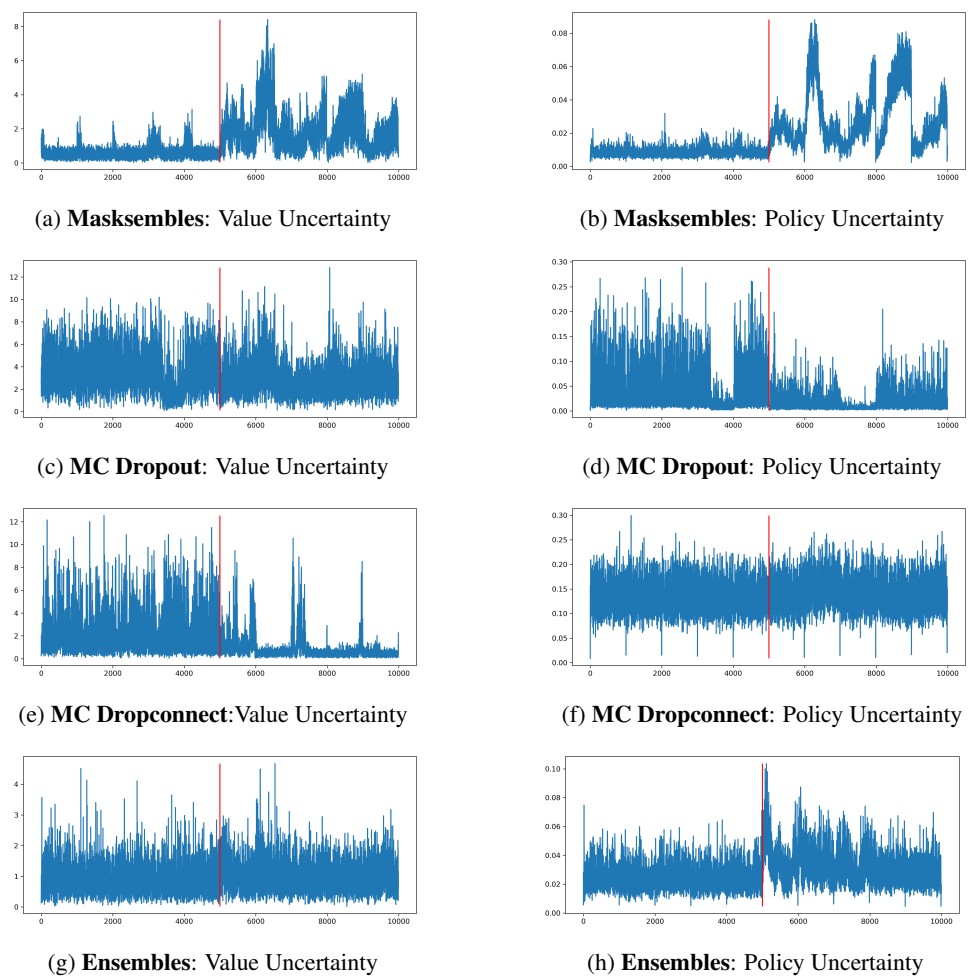

Figure 40: Value and policy uncertainty plots over time for different methods in *HalfCheetah-v3*. The first 5000 steps correspond to ID samples, the following 5000 samples – to OOD samples, separated by the vertical red line.

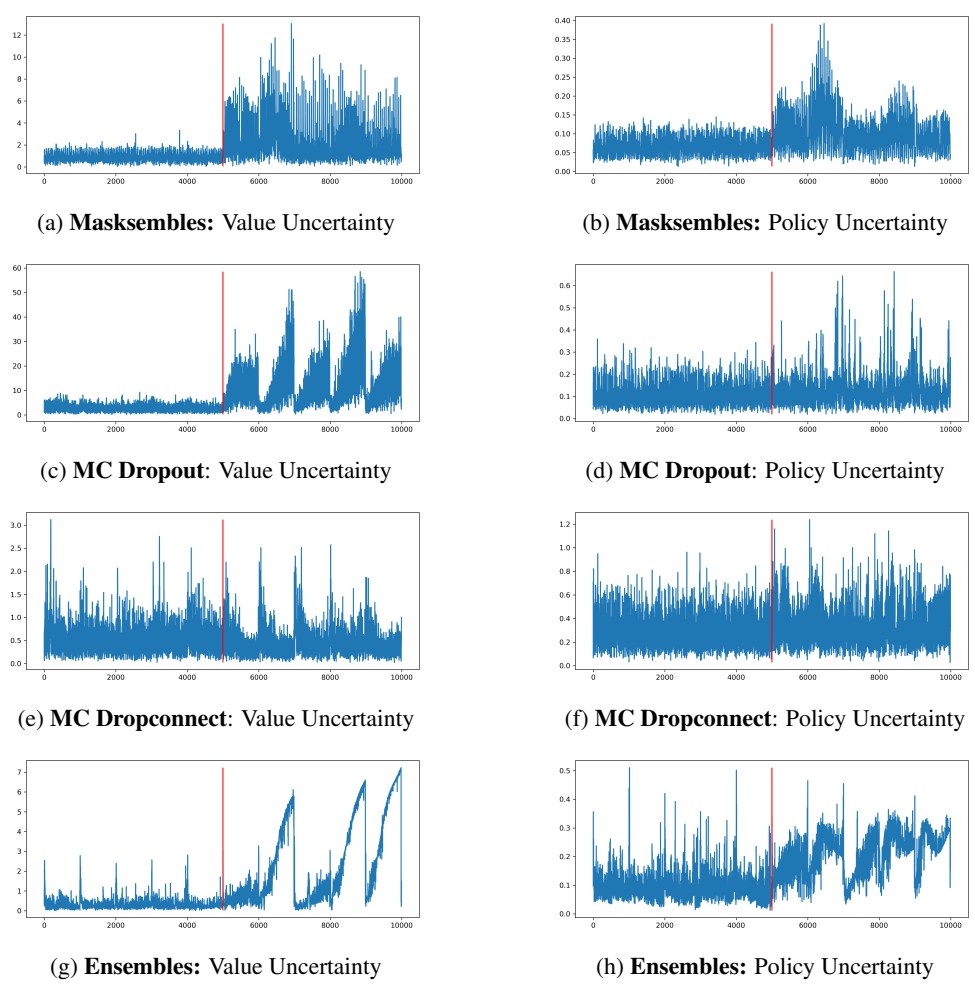

(a) **Masksembles:** Value Uncertainty

(b) **Masksembles:** Policy Uncertainty

(c) **MC Dropout**: Value Uncertainty

(d) **MC Dropout**: Policy Uncertainty

(e) **MC Dropconnect**: Value Uncertainty

(f) **MC Dropconnect**: Policy Uncertainty

(g) **Ensembles:** Value Uncertainty

(h) **Ensembles:** Policy Uncertainty

Figure 41: Value and policy uncertainty plots over time for different methods in *Swimmer-v3*. The first 5000 steps correspond to ID samples, the following 5000 samples – to OOD samples, separated by the vertical red line.

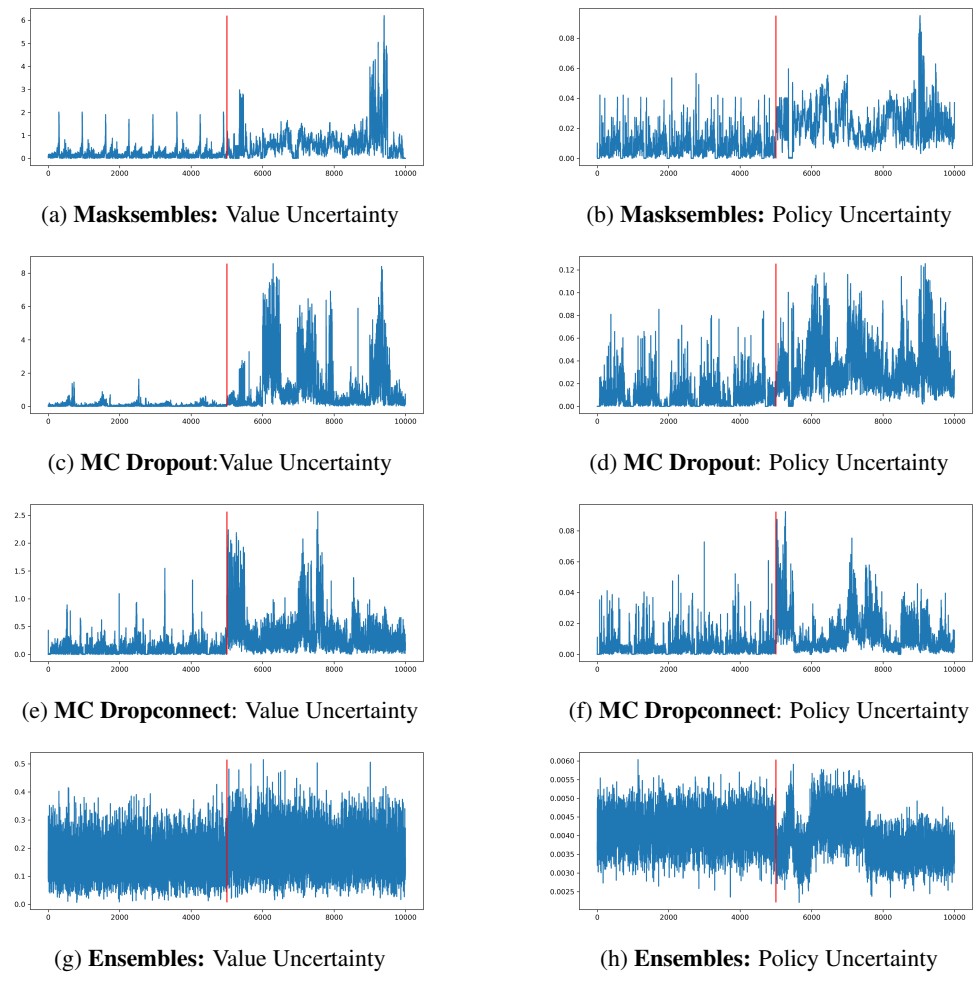

Figure 42: Value and policy uncertainty over time for different methods in *MsPacmanNoFrameskip-v4*. The first 5000 steps correspond to ID samples, the following 5000 samples – to OOD samples, separated by the vertical red line.

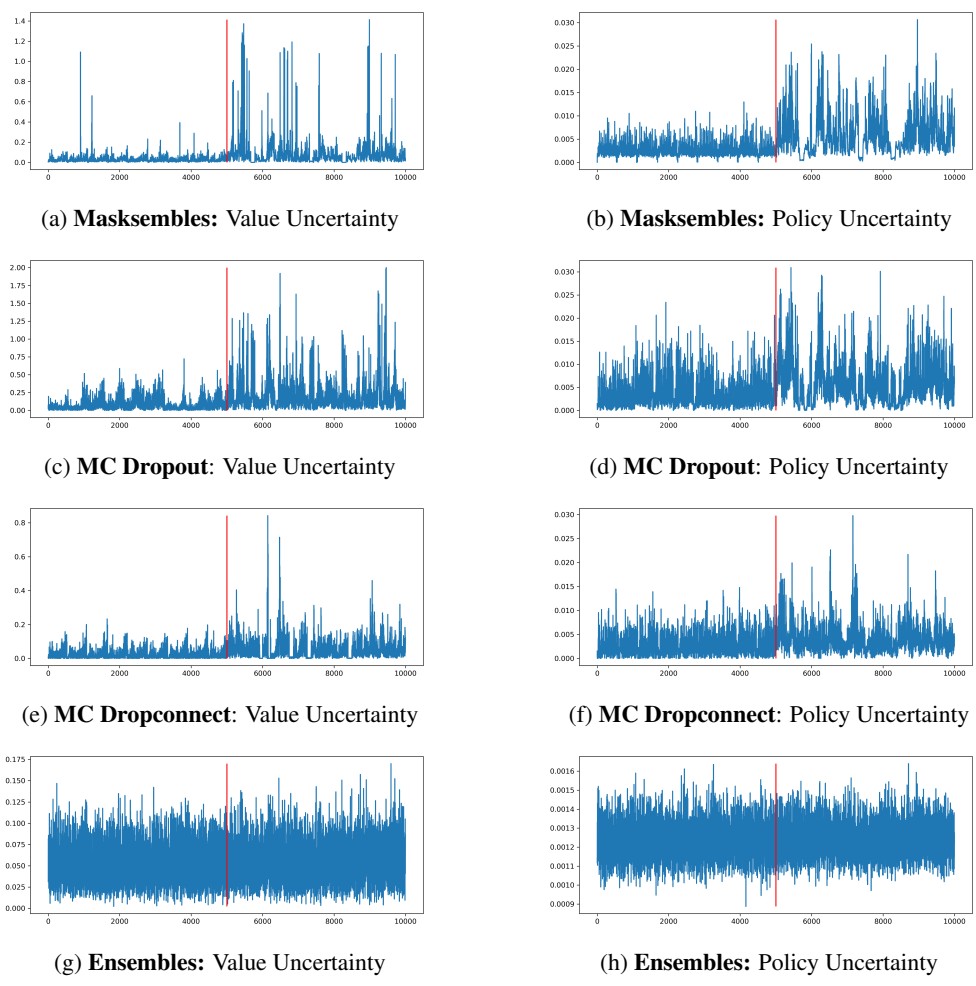

(a) **Masksembles:** Value Uncertainty

(b) **Masksembles:** Policy Uncertainty

(c) **MC Dropout**: Value Uncertainty

(d) **MC Dropout**: Policy Uncertainty

(e) **MC Dropconnect**: Value Uncertainty

(f) **MC Dropconnect**: Policy Uncertainty

(g) **Ensembles:** Value Uncertainty

(h) **Ensembles:** Policy Uncertainty

Figure 43: Value and policy uncertainty over time for different methods in *SeaquestNoFrameskip-v4*. The first 5000 steps correspond to ID samples, the following 5000 samples – to OOD samples, separated by the vertical red line.

