# OpenReview forum: "How to Enable Uncertainty Estimation in Proximal Policy Optimization"
_ICLR.cc/2023/Conference — Submitted to ICLR 2023_

### Official Review · Reviewer_5Wtm · 2022-10-17

**Confidence:** 4
**Correctness:** 2
**Technical Novelty And Significance:** 2
**Empirical Novelty And Significance:** 2
**Recommendation:** 3

**Clarity, Quality, Novelty And Reproducibility:**

**Clarity and Quality.** The clarity and the quality of the paper need to be further improved, as I mentioned above

**Novelty.** Using std of neural logits for representing uncertainty and adding Maskembles layers into the neural model define the main contributions in these works. These ideas are incremental and marginal, especially for the submissions to this top-tier cs conference.

**Reproducibility** Reproducibility should be satisfying, but my understanding could be biased since the paper is composed in a rush and is difficult to review.

**Strength And Weaknesses:**

### Strength

1. This paper explores an intriguing research topic about how to enable uncertainty in RL models.
2. The empirical studies demonstrate the leading performance of proposed models under multiple environments.

### Weaknesses

1. **Writing.** The writing style is cumbersome, including many incompatible expressions. I list some problems in the abstract, but there are more issues in the main paper. Reviewing based on the current version of this paper is difficult. It seems the paper introduces a fairly easy-to-follow model but the writing makes it hard to comprehend. Many details require further clarification. I strongly recommend using some help from the writing service from your school and other institution or companies in order to improve your writing.


To overcome these issues -> To resolve these issues

have not seen widespread adoption -> have not been extended to

To overcome the first gap -> To fill this gap

possible applicable -> potentially applicable

The second point -> The second issue or problem

We show experimentally -> Our experiment shows

among the survey methods -> among the previously-proposed method or in a survey paper?

detection while matching -> that matches

via a custom evaluation benchmark  -> via a customized evaluation benchmark

2. **Uncertainty Estimation.** It seems the authors misunderstand the concept of uncertainty in RL. An RL task commonly has two kinds of uncertainties, including epistemic uncertainty and aleatoric uncertainty. At the beginning of this paper, the authors emphasize the problem of the OoD dataset, which is a typical source of epistemic uncertainty, but then in the benchmark, the authors mention "perturbations to the parameters of the physics simulation", which, on the other hand, defines aleatoric uncertainty. I recommend looking into the definition of uncertainties in a common machine learning class. The works from Yarin Gal's group could serve as nice examples, for instance

Jishnu Mukhoti, Andreas Kirsch, Joost van Amersfoort, Philip H. S. Torr, and Yarin Gal.
Deterministic neural networks with appropriate inductive biases capture epistemic and aleatoric uncertainty. CoRR, abs/2102.11582, 2021.

3. **Comparison Methods.** The paper argues "In contrast, our work specifically targets for on-policy and actor-critic methods." I believe the uncertainty of Q values can define policy uncertainty since both Q-learning methods and policy gradient methods rely on the Q values for determining or updating policy. The uncertainty of cumulative rewards can directly capture the uncertainty in an RL task since the cumulative rewards term is the objective of RL agents. Compared to policy, it directly reflects the agent's preference under different states and is more compatible with the goal of MDP. More importantly, **I do not understand why we can not use the uncertainty of Q values to detect OoD samples.** What's the problem and why not compare with previous methods?

4. **Additional Issues**

- In Formula (3). The logits have not been normalized. Their ranges can significantly deviate from one to another. Their stds do not have a consistent meaning.

- In Figure 2. What does the dashed line mean in this plot?

- In section 5. What's the problem with using a D4RL dataset? We can create an OoD dataset by simply excluding the optimal trajectories or sub-optimal from the D4RL dataset.

- "Two-Objective Optimization" in Section 6. It seems the Maskembles method does not require an objective to optimize. It is in fact a neural architecture that can be integrated into any of the deep models. The deep model does not have to be for solving the RL problem. The only objective the paper optimized is the traditional RL objective.

- In Figure 3. The legends and labels are impossible to read. Please resize them.

**Summary Of The Paper:**

This paper introduces the approach of incorporating an uncertainty estimator into the PPO model. The authors define the uncertainty of RL models by the standard deviation of logits from neural models and propose capturing these uncertainties by utilizing a previously proposed model named Masksembles. To evaluate the model, this paper introduces a benchmark that includes OoD datasets in the previously studied RL environment, such MojoCo, and Atari. The empirical evaluation shows the RL model with Masksembles layers can outperform previous work in terms of collecting more rewards and detecting OoD samples.

**Summary Of The Review:**

This paper has some critical issues in writing. The contributions are marginal, and I have listed some concerns about their major contributions, especially about the concept of uncertainty. It seems the paper is composed in a rush. I recommend revising the paper for a reader-friendly format. I vote for a rejection based on the current format.

---

### Official Review · Reviewer_dCQj · 2022-10-25

**Confidence:** 3
**Correctness:** 3
**Technical Novelty And Significance:** 2
**Empirical Novelty And Significance:** 2
**Recommendation:** 5

**Clarity, Quality, Novelty And Reproducibility:**

## Clarity
The writting is clear.

## Quality
The quality is good.

## Novelty
The proposed definition of policy and value uncertainty for RL is novel.

## Reproducibility
The provided details should be sufficient for reproducibility.



**Strength And Weaknesses:**

## Strength

+ Introduce the concept of uncertainty fir on-policy actor-critic RL, specifically, introduce policy and value uncertainty in section 3. There is limited work study uncertainty for OOD detection in RL. Unlike iid samples in supervised learning, defining indistribution and out-of-distribution data for RL is challeging. Also the definition of uncertainty for RL should be different than that in supervised learning. Previous uncertainty definition in supervised learning does not scale to RL domains. This work smartly present the uncertainty of RL policy and value as the disagreement between submodels. The disagreement was performed as the standard deviation between submodels and alternatively as Jensen-Shannon divergence between output distributions (in Appendix D). With the definition of policy and value uncertainty, existing uncertainty estimation methods can be intergrated for uncertainty estimation in RL setups. Then, the OOD detection was framed as a binary classification problem, using the computed uncertainty scores as a threshold to classifiy ID and OOD states.

+ The empirical study was conducted in both discrete action space and continuous action space. The limitations of this work has been discussed.

## Weakness

+ The main contribution of this work is providing definitions for policy and value uncertainties in RL. However, the practical uncertainty estimation implementation directly uses existing uncertainty estimation methods. If a novel uncertainty estimation method can be proposed  for RL would further increase the quality of this work.




**Summary Of The Paper:**

This paper focues on quantify the uncertianty of online RL and detect out-of-distribution (OOD) states. Specifically, the definitions of uncertainty and OOD was presented for proximal policy optimization. The cocepts of value and policy uncertainty were discussed. Moreover,  this paper formulates a Pareto optimization problem to overcome the trade-off between reward and OOD detection. Empirically, they found Masksembles enables high-quality uncertainty estimiation and OOD detection while matching the performance of original RL agents.

**Summary Of The Review:**

Overall, I found this work is a intreresting work and novel in providing the definition of policy and value uncertainty.

---

### Official Review · Reviewer_vngt · 2022-10-26

**Confidence:** 3
**Correctness:** 3
**Technical Novelty And Significance:** 2
**Empirical Novelty And Significance:** 3
**Recommendation:** 6

**Clarity, Quality, Novelty And Reproducibility:**

The exposition in the paper was high quality and clear. Although there was not a particular focus in this paper on novelty, but rather framing and exploration of OOD detection in an RL setting, the authors brought up important points of discussion for this area of research and empirically studied a good number of baselines across many simulation settings. Aside from the points of clarification listed above, I found the paper to be clear. Code is provided in the supplemental material, which aids reproducibility.

**Strength And Weaknesses:**

Strengths:
* I really enjoyed reading this paper. It is well written and the ideas are easy to follow.
* The topic is timely in terms of OOD detection in RL for safety-critical applications.
* The problem is well-motivated and the literature review does a good job of contextualizing the paper in prior work.
* The methods section introduced concepts eloquently, clearly, and sequentially - excellent exposition!
* The figures are informative and effectively illustrate the benefits of the proposed approach.
* The experiments consider a good number of baselines and many simulation settings, as well as include a thorough ablation study.
* The future work and limitations sections were interesting and pertinent to the research community. Specifically, some important insights and open problems to address are:
    * The OOD detection and reward trade-off, which includes robust uncertainty estimation methods potentially not being performant for on-policy RL;
    * Training data is constantly changing for on-policy RL causing the OOD states to evolve and catastrophic forgetting causing ID states to become OOD;
    * Since there is not necessarily a singular correct action, a low probability for a certain action does not necessarily correspond to high epistemic uncertainty;
    * Framing of OOD makes sense for RL as valid game states that the agent has not yet encountered rather than, for example, random perturbations to images.

Weaknesses:
* The Pareto optimization problem is only explicitly mentioned in the abstract. I am assuming this is referring to the Fig. 4 discussion?
* The paper lacks polish as there are a number of formatting and referencing inconsistencies (see below).
* A more in-depth discussion of how the proposed approach differs from Clements et al. (2019) and Charpentier et al. (2022) would be prudent.
* The authors highlight that since there is not necessarily a best correct action in on-policy RL inducing high aleatoric uncertainty, this does not necessarily correspond to high epistemic uncertainty. Does this not mean that methods that disentangle aleatoric and epistemic uncertainty (e.g., Charpentier et al. 2022) would work better in the RL setting? Similarly, the discussion that disagreement between sub-models causes a decrease in reward appears to lead to the benefit of one-shot uncertainty estimation models, such as Charpentier's evidential deep learning architecture. It seems that baselining against such an approach would make the paper stronger.
* Since Masksembles are a critical part of the paper, it would be helpful to include a more detailed explanation of the method in Sec. 4.1.
* The performances in Tables 1 and 2 were not particularly decisive, although the Masksemble did seem to outperform in the majority of the settings. It would be helpful if standard errors across the 3 seeds were reported.
* I was missing a takeaway conclusion for the experiment in the rewards paragraph in Sec. 6. Why is it important to note that multiple predictions does not necessarily lead to better rewards?
* As the authors state, it is very surprising that ensembles performed so poorly in the PPO setting given many prior successes across domains. Do the authors know of any other papers that support poor performance of ensembles in on-policy settings? Perhaps 4 models in the ensemble is not sufficient?
* The experiments section would benefit from additional qualitative and quantitative analysis. Qualitatively, it would be interesting to look at some example states that were marked as OOD by Masksembles. Quantitatively, it would be helpful to look at perturbation magnitude (in continuous perturbation settings) versus estimated epistemic uncertainty for different methods to examine if an increase in perturbation leads to higher estimated uncertainty.
* There was no clearly better approach in terms of value versus policy uncertainty estimation and no associated discussion.

Some typos and points of confusion are listed below:
1. Sec. 1, paragraph 1: 'safety-critic' -> 'safety critical'.
2. The reference citation format is inconsistent throughout the paper, often with citations being placed with or without parentheses arbitrarily (e.g., first example is second line on page 2).
3. OOD and ID acronyms are not defined in the body of the paper (should be defined in paragraph 1 of page 2).
4. Sec. 2, paragraph 1: 'aleatoric' -> 'epistemic'.
5. 'Bayesian' should be capitalized.
6. Sec. 3, second paragraph: 'Markov decision process is set $\mathcal{M}$ is incorrect grammar.
7. Sec. 3, third paragraph: should all the $a$ and $s$ in the sentence 'Formally...' have $t$ subscripts?
8. Eqs. 1, 4, 5: missing periods at the end of sentences.
9. Eq. 2, I do not believe $\tilde{\pi}$ was defined formally.
10. Sec. 4.1, second paragraph: 'determine flexibly determine'.
11. Sec. 4.1, third paragraph: missing space in 'Fig.2'.
12. Fig. 2 caption seems like the semi-colon should be a period?
13. Sec. 5, second paragraph: 'varying the dimensions of body bodies'.
14. Sec. 5, second paragraph: 'These modifications create OOD states when the agent acts in it.' is a bit grammatically awkward.
15. Sec. 5: referencing 'section 6' instead of 'Sec. 6'.
16. Sec. 6, reward paragraph, line 2: extra parentheses added in two places.
17. Sec. 6, reward paragraph: 'in 4.2' -> 'in Sec. 4.2'.
18. Sec. 6, reward paragraph: 'in the Tab. 1' -> 'in Tab. 1'.
19. Tables 1 and 2 were not properly formatted. Try using the booktabs package for formatting tables.
20. MS, MCDO, MCDC acronyms were not defined in Table 1, but were defined in the caption of Table 2.
21: Sec. 6, OOD detection paragraph: 'These results' -> 'The results',
22: Fig. 4 axis labels are too close to the sub-figure labels.
23. Fig. 4 caption: 'The configuration are sampled' -> 'The configurations are sampled'.
24. Table 2: V. U./P. U./Value U. were not defined. 'MC DO', 'MC DC' have a space in this table, but not in Table 1.
25. Appendix should start on a new page after references list.
26. Appendix sections should consistently either start on a new page or not.
27. Appendix F, paragraph 2: extra space in 'Eq.  5'.
28. The references should be proofread (e.g., to ensure the year is not entered twice in a citation, the conference venue is listed instead of ArXiv when available, the conference name formatting is consistent, etc.).

**Summary Of The Paper:**

The paper presents a study into epistemic uncertainty estimation and OOD detection for reinforcement learning (RL), focusing on proximal policy optimization (PPO). The paper frames the OOD detection problem for RL, the differences and challenges encountered for RL as compared to supervised learning, and proposes value and policy based uncertainty measures. Several methods were considered for uncertainty estimation: Masksembles, MC Dropout, MC Dropconnect, and ensembles across many simulation settings. Masksembles was found to maintain the performance of RL (in terms of reward) while outperforming other methods in uncertainty estimation.

**Summary Of The Review:**

Overall, this is an interesting, clearly written paper that considers a timely problem of epistemic uncertainty estimation in on-policy RL. The authors do their due diligence in considering several baselines across many datasets. The paper frames important considerations for OOD detection in the RL setting. However, further insight gleaned from the experiments in terms of statistic significance and policy versus value uncertainty, as well as baselining against a one-shot uncertainty estimation method (e.g., Charpentier et al., 2022) would strengthen the paper.

---

### Official Review · Reviewer_rBt3 · 2022-10-26

**Confidence:** 5
**Correctness:** 2
**Technical Novelty And Significance:** 1
**Empirical Novelty And Significance:** 2
**Recommendation:** 3

**Clarity, Quality, Novelty And Reproducibility:**

The paper is very well written, but seems to get a lot of details wrong. The paper should also make clearer what the contributions are and what is already well known (like "We, therefore, hypothesize that we can interpret these average standard deviations as a measure of the agreement/disagreement", which already has been well evaluated). Although the extensive appendices make reproducibility appear very feasible, the paper has almost no novelty that would warrant publication.

**Strength And Weaknesses:**

**Stength**

- OOD detection, as a first step to react somehow to it, is an important topic in RL research.

**Weaknesses**

- Unless the reviewer misunderstood something fundamental, there seems to be almost no novelty here. The authors empirically compare known uncertainty estimators, most of them known to work well for OOD in RL (see missing references below), and come to the conclusion that the newest method (Masksembles) works the best. The only other novelty is how OOD samples are generated, which did not convince this reviewer (see below).
- The evaluation is in this reviewers eyes insufficient and conceptually flawed. First, every uncertainty method seems to reduce the performance. Why is that? Second, none of the tables report standard deviations over the 3 seeds, which are very few to make a strong argument anyway. And last, the way the OOD data is generated is very random and does not allow to say how similar evaluated samples are to the training data. Some samples will therefore be very similar and others dissimilar, without a way to correct for this confounding factor.
- The authors also seem to misunderstand epistemic and aleatoric uncertainty. Most of Section 3.2 concerns methods that estimate the *aleatoric uncertainty* of $\pi$, which does not affect OOD! For example, the confidence or entropy of the policy network can be extrapolated to be extremely certain, even if the state is OOD. Only the ensemble methods (including dropout) estimate actually estimate the required epistemic uncertainty.
- The definition of policy uncertainty remains questionable: in RL multiple policies can have the same (or near identical) values. The same problem can therefore have multiple optimal policies that assign different probabilities to the same actions. Due to the training procedure this will probably not happen very often, but it still makes the metrics somewhat questionable.
- The introduction of the evaluated methods omits many (possibly important) details. Furthermore, MC Dropconnect is neither defined nor cited in the main paper.
- The paper does not cite a large body of works on exploration in RL that uses OOD detection with ensembles (and other methods) to produce intrinsic rewards (see below).
- Finally, the reviewer is not certain what the takeaway from the paper is. The authors present two OOD detectors based on value and policy, but how should those be used and what is the meaning if they should disagree? To warrant publication, the authors should introduce some novel idea or analysis and demonstrate them on tasks that are of practical interest, like e.g. exploration.

**Missing references**

Marc G. Bellemare, Sriram Srinivasan, Georg Ostrovski, Tom Schaul, David Saxton, and Rémi Munos. "Unifying count-based exploration and intrinsic motivation." In Advances in Neural Information Processing Systems (NIPS) 29, pages 1471-1479, 2016. URL https://arxiv.org/abs/1606.01868

Yuri Burda, Harrison Edwards, Amos J. Storkey, and Oleg Klimov. "Exploration by random network distillation." In 7th International Conference on Learning Representations (ICLR), 2019. URL https://openreview.net/forum?id=H1lJJnR5Ym

Meire Fortunato, Mohammad Gheshlaghi Azar, Bilal Piot, Jacob Menick, Matteo Hessel, Ian Osband, Alex Graves, Volodymyr Mnih, Remi Munos, Demis Hassabis, Olivier Pietquin, Charles Blundell, and Shane Legg. "Noisy networks for exploration." In International Conference on Learning Representations (ICLR), 2018. URL https://arxiv.org/abs/1706.10295

Xiuyuan Lu and Benjamin Van Roy. "Ensemble sampling." In Advances in Neural Information Processing Systems (NeurIPS), volume 30, 2017. URL https://papers.nips.cc/paper/2017/hash/49ad23d1ec9fa4bd8d77d02681df5cfa-Abstract.html

Brendan O’Donoghue, Ian Osband, Rémi Munos, and Volodymyr Mnih. "The uncertainty Bellman equation and exploration." In Proceedings of the 35th International Conference on Machine Learning (ICML), pages 3836–3845, 2018. URL https://arxiv.org/abs/1709.05380

Ian Osband, Benjamin Van Roy, and Zheng Wen. "Generalization and exploration via randomized value functions." In Proceedings of the 33rd International Conference on International Conference on Machine Learning (ICML), pages 2377–2386, 2016. URL https://arxiv.org/abs/1402.0635

Ian Osband, John Aslanides, and Albin Cassirer. "Randomized prior functions for deep reinforcement learning." In Advances in Neural Information Processing Systems (NeurIPS) 31, pages 8617–8629. 2018. URL https://arxiv.org/abs/1806.03335

Georg Ostrovski, Marc G. Bellemare, Aäron van den Oord, and Rémi Munos. "Count-based exploration with neural density models." In Proceedings of the 34th International Conference on Machine Learning (ICML), pages 2721–2730, 2017. URL https://arxiv.org/abs/1703.01310

Deepak Pathak, Pulkit Agrawal, Alexei A. Efros, and Trevor Darrell. "Curiosity-driven exploration by self-supervised prediction." In Proceedings of the 34th International Conference on Machine Learning (ICML), 2017. URL https://arxiv.org/abs/1705.05363

Haoran Tang, Rein Houthooft, Davis Foote, Adam Stooke, OpenAI Xi Chen, Yan Duan, John Schulman, Filip DeTurck, and Pieter Abbeel. "#Exploration: A study of count-based exploration for deep reinforcement learning." In Advances in Neural Information Processing Systems (NIPS) 30, pages 2753–2762. 2017. URL https://arxiv.org/abs/1611.04717

M. Sensoy, L. Kaplan, and M. Kandemir, "Evidential deep learning to quantify classification
uncertainty." arXiv preprint arXiv:1806.01768, 2018.

**Summary Of The Paper:**

The paper investigates OOD detection, with different ensembles amd dropout methods, of policy and value networks in PPO. Experiments report RL-performance and OOD-ROC curves on 4 Mujoco and 4 Atari benchmarks. All methods decreased perrformance in (almost all) environments, but Masksembles have been reported to work somewhat better for OOD.

**Summary Of The Review:**

The paper does, in the eyes of this reviewer, not have enough novelty to warrant publication. While well written, the paper misrepresents some key aspects and does not yield a particularly actionable result. The reviewer therefore strongly argues for rejection.

To change this recommendation, the authors would have to convince the reviewer that there is significant novelty in the paper that has been overlooked in this review.

---

### Decision · Program_Chairs · 2023-01-20

**Decision:**

Reject

**Justification For Why Not Higher Score:**

* Lack of novelty (uncertainty and OOD techniques already exist)
* Confusion about epistemic and aleatoric uncertainty
* Insufficient empirical evaluation


**Justification For Why Not Lower Score:**

NA

**Metareview: Summary, Strengths And Weaknesses:**

The paper studies uncertainty quantification and out-of-distribution state detection in RL.

Strengths:
* Timely and important topic
* Well-written paper

Weaknesses:
* Lack of novelty (uncertainty and OOD techniques already exist)
* Confusion about epistemic and aleatoric uncertainty
* Insufficient empirical evaluation

The paper does not propose a new technique, but it focuses on studying existing uncertainty and OOD techniques in the context of RL.  Given the insufficient empirical evaluation and the confusion about some aspects of epistemic and aleatoric uncertainty, this work is not ready for publication.